# MADA-Attack: Transferable Multi-modal Attention Distraction Adversarial Attack against Vision Language Models

**Zhihan Qin** [* 1]   **Jiahao Chen** [* 2]   **Chunyi Zhou** [2]   **Yuwen Pu** [1]   **Chunqiang Hu** [1]   **Xiaolei Liu** [3]   **Shouling Ji** [2]

## Abstract

Vision Language Models (VLMs) achieve strong performance across multi-modal tasks but remain vulnerable to universal adversarial perturbations (UAPs). Existing UAP methods mainly operate on the visual modality, overlooking structured textual semantics and cross-modal interactions, which limits their ability to disrupt alignment and generalize across tasks and model architectures. To address these limits, we propose **Multi-modal Attention Distraction Adversarial Attack (MADA-Attack)** framework. We begin by conducting several insight experiments and discover that modality attention distributes differently over layers and early phase of optimization is decisive. Building on these observations, we introduce Semantic Token Manipulation (STM) to steer text-guided attention, and Fused Embedding Training (FET) to jointly optimize textual and visual embedding losses for coordinated misalignment. We further incorporate an Adaptive Data Augmentation (ADA) strategy that dynamically balances attack strength, transferability, and training efficiency. Extensive experiments demonstrate that **MADA-Attack** consistently achieves state-of-the-art performance and strong transferability while remaining computationally lightweight, with an average ASR of 82.60% and 73.42% in zero-shot classification and image captioning tasks. For the visual question answering (VQA) and I-T Retrieval task, our method exceeds the SOTA baseline by 10%. Our code is available at this GitHub Repository.

---

[*]Equal contribution   [1]School of Big Data and Software Engineering, Chongqing University, Chongqing, China [2]School of Computer Science and Technology, Zhejiang University, Zhejiang, China [3]China Academy of Engineering Physics, China. Correspondence to: Yuwen Pu <yw.pu@cqu.edu.cn>, Chunqiang Hu <chu@cqu.edu.cn>.

*Proceedings of the $43^{rd}$ International Conference on Machine Learning*, Seoul, South Korea. PMLR 306, 2026. Copyright 2026 by the author(s).

## 1. Introduction

As Contrastive Language–Image Pre-training (CLIP) (Radford et al., 2021) has demonstrated remarkable Zero-Shot (ZS) generalization, Vision-Language Pre-training (VLP) models that utilize the CLIP model as image encoder have been adopted in numerous downstream applications, including image captioning (IC), ZS classification, Visual Question Answering (VQA), and Image–Text retrieval (I-T Retrieval) (Mokady et al., 2021; Saha et al., 2024; Huynh et al., 2025; Zhao et al., 2023). Such success also enabled the prosperity of Vision Language Models (VLMs), such as BLIP2 (Li et al., 2023), LLaVA (Liu et al., 2023a), MiniGPT4 (Zhu et al., 2023), and OpenFlamingo (Awadalla et al., 2023).

However, VLMs also exhibit inherent vulnerability toward adversarial attack (Goodfellow et al., 2014; Madry et al., 2017; Zhao et al., 2023; Xie et al., 2025). Among existing attack paradigms, Universal Adversarial Perturbation (UAP) poses a concerning challenge (Zhang et al., 2024). A single, imperceptible perturbation can mislead a model across diverse tasks, making UAP a scalable and practical attack method (Zhang et al., 2021; Liu et al., 2023b). UAPs also support transfer attacks, enabling cross-model, cross-architecture, and cross-task adversarial transferability without requiring access to models' gradients, an essential property for real-world adversarial evaluations (Chen et al., 2025; Qin et al., 2022; Wang et al., 2024; Chen et al., 2023). Consequently, CLIP-based systems face increasing security risks during practical deployments and raise public concern.

Recent work has shifted toward improving the transferability of UAPs in VLMs. Most existing approaches formulate the problem as image-only optimization, where perturbations are learned solely from visual signals to degrade downstream predictions (Huang et al., 2025; Weng et al., 2024). However, such designs are limited to modality-specific visual distortions and neglect the role of cross-modal semantic alignment, which is fundamental to the generalization ability of modern VLMs (Huang et al., 2025; Schlarmann et al., 2024; Zhou et al., 2023). To address this gap, several recent studies proposed to incorporate textual information into UAP generation (Zhang et al., 2025a; Fang et al., 2025; Zhang et al., 2024; Wang et al., 2024). However, these methods are developed and evaluated for I–T

retrieval, leaving their effectiveness and generality across other vision–language tasks underexplored. Moreover, the cross-modal interactions in these approaches are typically introduced at the word or prompt level, which is less effective than embedding-level manipulation in disrupting cross-modal alignment (Zhang et al., 2025a), thereby hindering attack generalization across models and tasks.

To motivate our approach, we first analyze the limitations of existing multi-modal adversarial attacks through layer-wise attention analysis, revealing that fragmented cross-modal interactions fail to disrupt semantic alignment and limit transferability. We further examine the optimization dynamics of UAPs, observing a stability–transferability trade-off in which early training stages determine global perturbation structure, while later stages focus on local refinements.

Based on these insights, we propose **Multi-modal Attention Distraction Adversarial Attack (MADA-Attack)**, the first framework to explore **text-guided** non-targeted UAPs for VLMs. Our method introduces embedding-level textual guidance to steer UAP optimization in a shared embedding space. We further employ adaptive training strategies to stabilize early optimization while preserving semantic impact. Together, these components enable the learning of universal perturbations with improved semantic-level transferability across diverse VLMs and downstream tasks. Our main contributions are summarized as follows:

- We are the first to study text-guided non-targeted UAPs for VLMs and reveal the underlying mechanism between cross-modal alignment and UAP formation. Our analyses uncover the joint influence of textual and visual pathways on UAP formation and reveal multi-modal attention distraction pattern that first breaks benign alignment and then reconnects adversarial visual and textual representations.

- We propose **MADA-Attack** which enhances UAP optimization via embedding-level textual guidance. integrating **Semantic Token Manipulation (STM)** to model and disrupt the semantic latent distribution of text embeddings, **Adaptive Data Augmentation (ADA)** to balance attack stability and transferability through dynamic augmentation scheduling, and **Fused Embedding Training (FET)** to jointly optimize cross-modal perturbations by propagating adversarial textual semantics into visual embeddings across layers.

- **MADA-Attack** achieves black-box SOTA adversarial transferability across diverse tasks, with an average ASR of 82.60% on ZS classification (vs. 70.83% SOTA), 43.95% on IC task across three models, and leading results on VQA and I-T Retrieval benchmarks, demonstrating the effectiveness and generality of joint cross-modal optimization.

## 2. Related Work

### 2.1. Transferable Adversarial Attack

Due to its practical feasibility in black-box settings, transferable attacks have attracted increasing attention, where attackers have no access to the target model's parameters or architecture (Wang et al., 2021; Ma et al., 2023; Zou et al., 2023; Zhang et al., 2023; Huang & Kong, 2022; Qin et al., 2022; Ming et al., 2024). Existing methods for improving adversarial transferability can be categorized into three groups: transformation-based, gradient-based, and model-ensembling attacks. **Transformation-based** approaches employ data augmentations to prevent overfitting to surrogate models, with techniques such as random padding and resizing (Xie et al., 2019) shown to enhance robustness and transferability (Zhu et al., 2024; Wu et al., 2021). **Gradient-based** attacks optimize adversarial examples via iterative gradient updates, often building upon Projected Gradient Descent (PGD) (Madry et al., 2017). Representative variants improve transferability by incorporating momentum (Dong et al., 2018a), adaptive gradient directions (Yuan et al., 2024), or dynamic step sizes (Gao et al., 2020). **Model-ensembling** strategies improve transferability by jointly optimizing adversarial perturbations across multiple surrogate models. Recent studies explore ensemble loss optimization (Dong et al., 2018a), selective surrogate model sampling to reduce computational cost (Huang et al., 2025), and aim-oriented ensemble designs (Chen et al., 2023).

### 2.2. Multi-Modal Adversarial Attack

Building on advances in transferable attacks, recent research has shifted focus to multimodal adversarial attacks, exploring perturbations across both visual and textual domains. ICSA (Yin et al., 2023) provides cross-searching perturbations on different modalities, enabling its cross-tasks transferability. To better understand the difference between uni-modal and multi-modal attacks, Co-Attack (Zhang et al., 2022) explores how textual and visual modality interact with each other, and collaborate two modalities to concentrate on the same optimization goal, achieving better attack performance. MSI-Attack (Zhang et al., 2025a) further introduces the modality-specific and cross-modality features, undermining important embedding-aligned pixels, achieving high attack performance under strict perturbation budgets that are imperceptible to humans. To address the cross-modality interaction, C-PGC (Fang et al., 2025) utilize CLIP alignment mechanism, designing an interactive attack framework that disrupts image-text alignment and forces adversarial samples to move away from their original area in the VLM's feature space, hence achieving better attack performance.

Within the extensive body of research on multi-modal adversarial attack, the challenges are summarized as: **Cross-modal interaction** needs to be carefully designed as its

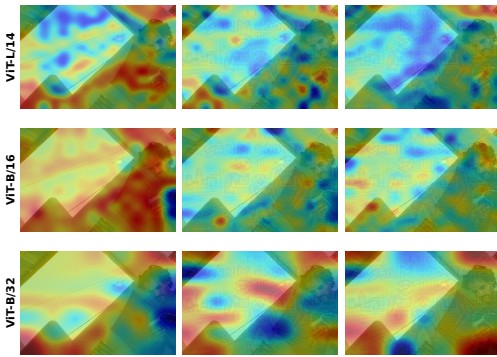

*Figure 1.* Visualization of the cross-modal alignment across CLIP models. From left to right, each column represents a clean image-text pair, a clean text and adversarial image pair and an adversarial image-text pair. The clean and adversarial texts are "*A man is reading a small book*" and "*the cover of the album*", respectively.

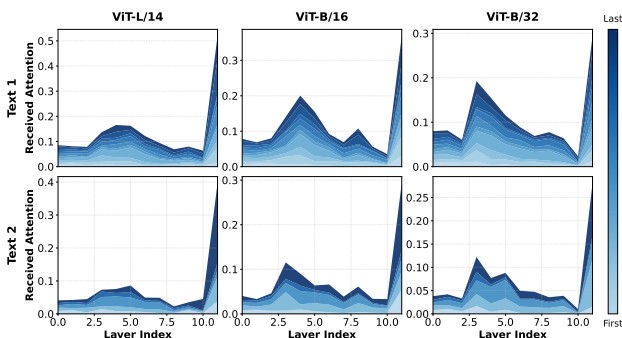

*Figure 2.* Attention distribution for text modality over layers across different models.

prominence for textual and visual optimization toward the same direction without mutual interference. **Modality embedding gap** needs mitigation that eventually contributes to semantic level disruption, which guarantees the transferability of such attack methods. In our work, we focus on how the textual and visual cues react differently across models.

## 3. Methodology

### 3.1. Design Intuition

**Textual Information Matters.** Recent studies indicates that VLMs are dominated by textual guidance, which can override visual evidence and induce incorrect predictions under misleading or incomplete contexts (Liu et al., 2025). Moreover, CLIP-based models often miss salient visual regions due to irrelevant token generation during visual encoding (Li et al., 2025a), further demonstrating how token-level misalignment undermines multimodal integration. Motivated by these findings, we argue that incorporating textual information to disrupt cross-modal interpretation at the embedding level is critical for transferable adversarial attacks. To further examine the roles of text and image modalities, we conduct an exploratory experiment and visualize the cross-attention patterns in Figure 1. We observe that adversarial examples disrupt the alignment between adversarial images and clean text by dispersing cross-attention, shifting it from semantically corresponding regions to scattered ones. Across different architectures, cross-attention further exhibits a re-alignment tendency toward text-referenced regions, suggesting that semantic alignment disruption is a universal and model-agnostic phenomenon.

**From Point-wise to Distributional Robustness.** Existing adversarial attacks often optimize against a fixed text prompt (Yin et al., 2024; Lu et al., 2023), which is susceptible to overfitting the specific linguistic features or the tokenization patterns of the surrogate model. While we ar-

gue that, since VLMs are trained to align visual and textual modalities within a high-dimensional continuous manifold, a singular prompt represents only a discrete point on this manifold. By constructing a semantic latent distribution $\mathcal{P}_T(t)$ given a text prompt $t$, we find the "semantic core" of a category, ensuring that the adversarial perturbation remains effective against any variant of the textual description, achieving semantic *Distributional Robustness*. To support this intuition, we analyze layer-wise attention patterns induced by different textual cues. We compare two text structures: Text 1, "*A man looks at a book on a sunny day*", and Text 2, "*A photo of a cat*". As shown in Figure 2, textual attention increases across layers in multiple models. Moreover, detailed descriptions lead to dispersed attention, whereas concise and specific text results in attention merging toward the final semantic token. This behavior indicates that classification-style prompts encourage models to focus on dominant objects and a single prompt is not enough to capture the semantic distribution of the image.

### 3.2. MADA-Attack

**Semantic Token Manipulation.** To incorporate textual information into UAP generation, we introduce STM. STM initially adopts a text template "***A photo of [CLS]***", where [CLS] denotes the class token associated with each image. Specifically, given a surrogate image encoder $f_I$ and text encoder $f_T$, the image UAP $\delta^I$ optimization over the retained dataset $\mathcal{D}$ is formalized as:

$$\max_{\delta^I} \ \mathbb{E}_{(x_i,t_i)\sim\mathcal{D}} \left[ \mathrm{dist} \left( f_I(x_i + \delta^I), \ f_T(t_i) \right) \right], \quad (1)$$

where $t_i$ represents a single, deterministic text prompt of the corresponding image $x_i$. However, optimizing against a discrete point in the embedding space often leads to overfitting the specific linguistic patterns of the surrogate text encoder, as stated before, limiting adversarial transferability.

To address this, we shift the optimization pipeline from point-wise minimization to the disruption of the entire *semantic latent distribution* $\mathcal{P}_T$ associated with $t_i$. In modern VLMs, the final token embedding of the text encoder

compresses the global semantic information of the input sequence and is the primary representative for cross-modal alignment (Radford et al., 2021). Given that text encoders naturally map semantically similar phrases into proximity within the embedding space, we characterize the semantic distribution $\mathcal{P}_T(t_i) = \{\tilde{e}_i^t \in \mathbb{R}^d | \|\tilde{e}_i^t - f_T(t_i)\| \leq \epsilon_t\}$ by sampling within an $\epsilon_t$-bounded neighborhood around the original text embedding $e_i^t = f_T(t_i)$. Here we use $f_T(t_i)$ to stand for the last token embedding of the text $t_i$. Consequently, the UAP optimization pipeline is reformulated to minimize the expected similarity from this distribution:

$$\max_{\delta^I} \mathbb{E}_{(x_i,t_i)\sim\mathcal{D}} \left[ \mathbb{E}_{\tilde{e}_i^t \sim \mathcal{P}_T(t_i)} \left[ \text{dist}\left(f_I(x_i + \delta^I), \tilde{e}_i^t\right)\right]\right]. \tag{2}$$

Since sampling directly from the continuous latent space is computationally prohibitive, we follow the principles of Distributionally Robust Optimization (Kuhn et al., 2025) and approximate the inner expectation by its lower bound. This transforms the objective into a double optimization problem, where we simultaneously seek the most representative semantic boundary via a learnable text perturbation $\delta^T$ and update the image UAP $\delta^I$ to maximize the misalignment:

$$\max_{\delta^I} \min_{\delta^T} \mathbb{E}_{(x_i,t_i)\sim\mathcal{D}} \left[ \text{dist} \left\langle f_I(x_i + \delta^I), \ f_T(t_i) + \delta^T \right\rangle \right],$$
$$\text{s.t. } \|\delta^I\| \leq \epsilon_i, \|\delta^T\| \leq \epsilon_t. \tag{3}$$

For efficiency consideration, we do not optimize perturbation for different $t_i$ and seek for universal textual perturbation $\delta^T$ to estimate the lower bound of the original expectation over $\mathcal{P}_T$. This distributional approach ensures that the resulting UAP $\delta^I$ is robust to semantic variations, significantly enhancing its transferability across diverse model architectures and downstream tasks. For detail implementations, we use cosine similarity measurement to replace dist($\cdot$). Following Huang et al. (2025); Fang et al. (2025), we use $\ell_\infty$ to obtain effective yet controllable perturbation.

**Fused Embedding Training.** With established framework above, we propose FET as a principled module to solve the optimization of Equation 3. FET is explicitly built upon the STM module, inheriting its carefully designed textual perturbations and propagating them into the visual domain through joint embedding-level optimization. To ground our design, we analyze the layer-wise attention allocation between image and text tokens across multiple CLIP-style models. As shown in Figure 3, image tokens allocate most of their attention to the `[CLS]` token, while their overall attention contribution decreases as layers go deeper. In contrast, textual tokens exhibit an inverse trend, with attention weights increasing in deeper layers. This complementary behavior suggests a consistent modality transition: visual tokens dominate shallow layers, whereas textual semantics increasingly govern deeper representations. Such non-simultaneous dominance motivates explicitly coupling image perturbation

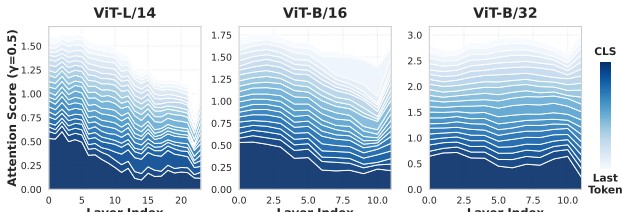

*Figure 3.* Attention distribution for image modality over layers and different models. From left to right represents ViT-L/14, ViT-B/16 and ViT-B/32, respectively.

with textual semantics during optimization.

Specifically, the optimization of FET is decomposed into an *Interaction Phase* and a *Fusion Phase*. In the Interaction Phase, we first solve for the optimal universal textual perturbation $\delta^T$ to approximate the worst-case semantic shift within $\mathcal{P}_T$. Given the complementary attention dynamics observed across layers, this phase explicitly targets the deep-layer semantic representations where textual influence is most pronounced. The textual perturbation is updated to minimize the distance from the adversarial image embeddings $\tilde{e}_i^x$ across the surrogate ensemble:

$$\delta_{j+1}^T = \delta_j^T - \eta\nabla_{\delta^T} \text{dist}(f_I(x_i + \delta^I), f_T(t_i) + \delta_j^T). \tag{4}$$

In the subsequent Fusion Phase, we propagate this semantic disruption into the visual modality. We define a joint representation space by fusing the adversarial textual embeddings with the perturbed image features. Let $u_c = \mathcal{F}(f_I(x_i), f_T(t_i))$ represent the benign fused embedding and $v_{adv} = \mathcal{F}(f_I(x_i + \delta^I), f_T(t_i) + \delta^T)$ represent the adversarial fused embedding, where $\mathcal{F}(\cdot)$ is an additive fusion operator. The image UAP $\delta^I$ is then updated to minimize the semantic alignment between $u_c$ and $v_{adv}$:

$$\delta_{j+1}^I = \Pi_{\epsilon_i} \left( \delta_j^I + \eta\nabla_{\delta^I} \mathbb{E}_{\mathcal{D}} \left[ \text{dist}(u_c, v_{adv})\right]\right). \tag{5}$$

To overcome the flat regions in the loss landscape often encountered in optimization, we incorporate momentum into the gradient updates for $\delta^I$ (Dong et al., 2018b).

**Adaptive Data Augmentation.** Recent studies have shown that UAPs tend to form repetitive chessboard-like artifacts, which significantly limit adversarial transferability across models and tasks (Liang & Pun, 2025). Incorporating data augmentation has been demonstrated to alleviate such artifacts by encouraging perturbations to encode more semantically meaningful patterns. Therefore, we introduce ADA to improve the robustness and transferability of UAPs. We conduct an analysis of the UAP optimization process as illustrated in Figure 5, which demonstrates that the early stage plays a decisive role in shaping the global structure of the perturbation, while later iterations refine local regions into coherent semantic patterns. This indicates that

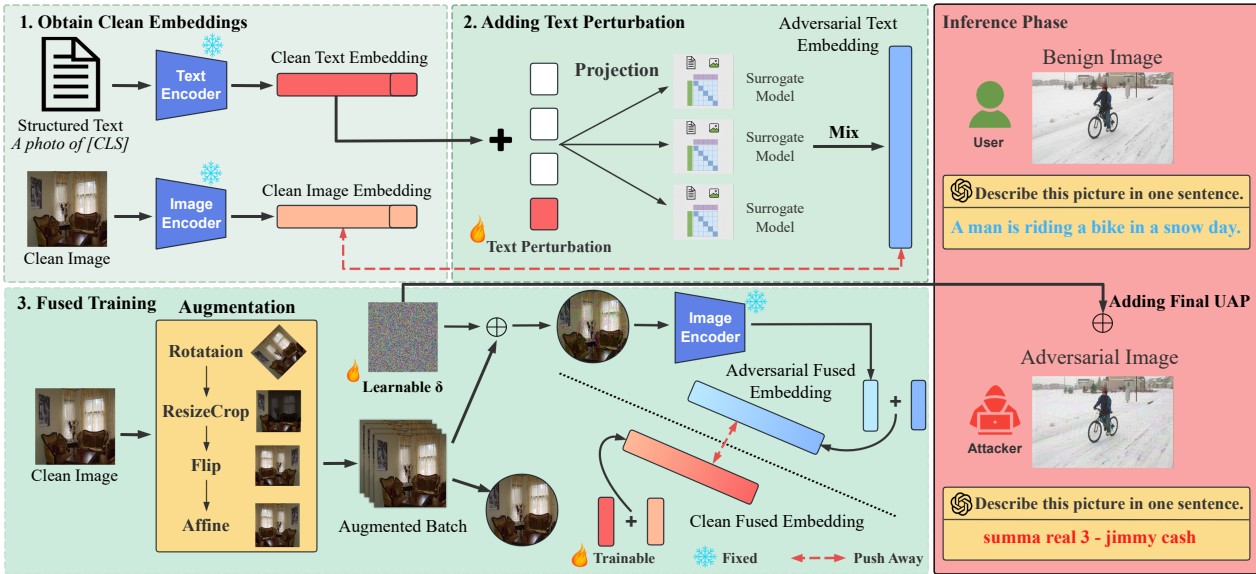

*Figure 4.* The pipeline of **MADA-Attack**, a unified UAP framework that leverages textual information to guide UAP optimization, disrupting cross-modal semantic alignment and enabling transferable attacks.

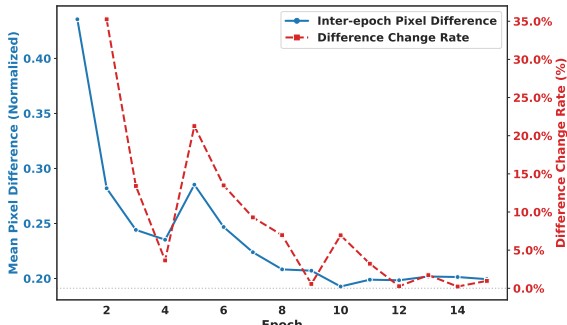

*Figure 5.* UAP mean change difference and pixel change rate.

the early phase determines the overall perturbation regularization, suggesting that aggressive data augmentation at early iterations may hinder the formation of stable semantic structures. Based on this insight, ADA departs from conventional fixed-probability augmentation strategies and instead adopts a training-aware augmentation schedule. We define an adaptive augmentation operator $\mathcal{A}_t(\cdot)$ parameterized by the optimization step $t$, where the augmentation probability increases monotonically over training: $\mathcal{A}_t(\cdot) \sim \mathcal{P}(\alpha_t), \alpha_t \uparrow$ as $t \to J$, with $\alpha_t$ denoting the augmentation strength or application probability at step $t$, and $J$ the total number of optimization steps. In practice, this design enforces low augmentation probability during early training to stabilize global perturbation formation, while increasing augmentation strength in later stages.

Following prior work on adversarial data augmentation (Zhu et al., 2024), we construct the augmentation pool using stan-

dard geometric transformations, including rotation, random cropping, horizontal flipping, and affine transformations. At each iteration, ADA dynamically samples a composition of these transformations according to the current augmentation probability $\alpha_t$. The full algorithm is provided in Appendix A. The effect of different augmentation sequences and schedules is further analyzed in Appendix B and a detailed implementation in Appendix E. The overview of our framework is shown in Figure 4.

## 4. Experiments

### 4.1. Experiment Settings

**Baseline.** We compare our method against a broad range of SOTA UAP approaches, including C-PGC (Fang et al., 2025), ETU (Zhang et al., 2024), AdvCLIP (Zhou et al., 2023), TRM-UAP (Liu et al., 2023c), GD-UAP (Mopuri et al., 2018), Meta-UAP (Weng et al., 2024), and XTransfer (Huang et al., 2025). For XTransfer, we directly adopt the official checkpoints provided in XTransferBench (Huang et al., 2025) for inference. In particular, we include both XTransfer Naive and XTransfer Base for comparison, using the official UAPs that ensemble one model and four models, respectively. These UAP methods span diverse target models and tasks, which enables a comprehensive evaluation.

**Evaluation Metrics.** Following Huang et al. (2025), we evaluate performance using ASR. Specifically, we compute the similarity between the model outputs and the reference output for both clean and adversarial inputs, denoted as $s_{\text{clean}}$

*Table 1.* Non-targeted ASR (%) results on the image captioning task across different VLMs on the Flickr30K dataset. Higher results indicate better attack performance. The second-best results are underlined and the best results are **boldfaced**.

| Model | Method | BLUE-1 | BLUE-2 | BLEU-3 | BLEU-4 | METEOR | ROUGE-L | CIDEr |
|---|---|---|---|---|---|---|---|---|
| MiniGPT4 | AdvCLIP | 17.06 | 21.50 | 24.42 | 26.35 | 20.61 | 21.07 | 57.28 |
| | ETU | 18.81 | 24.22 | 27.62 | 30.18 | 26.81 | 26.11 | 61.51 |
| | Meta-UAP | 19.87 | 25.89 | 29.98 | 33.21 | 23.59 | 23.80 | 60.50 |
| | TRM-UAP | 18.71 | 23.82 | 27.58 | 30.54 | 24.60 | 25.06 | 61.18 |
| | GD-UAP | 20.85 | 26.39 | 29.99 | 32.78 | 31.54 | 30.67 | 64.50 |
| | C-PGC | 22.11 | 27.64 | 31.21 | 33.95 | 34.99 | 33.83 | 67.51 |
| | XTransfer Naive | 19.85 | 25.35 | 29.18 | 32.09 | 29.33 | 28.85 | 62.78 |
| | XTransfer Base | 24.15 | 30.80 | 34.96 | 38.33 | 38.21 | 36.27 | 68.05 |
| | **MADA-Attack(Ours)** | **28.76** | **34.90** | **38.33** | **40.93** | **47.11** | **44.65** | **72.67** |
| BLIP2 | AdvCLIP | -5.89 | -2.00 | 1.87 | 4.57 | 0.14 | 8.70 | 6.81 |
| | ETU | -4.59 | -0.10 | 4.28 | 7.65 | 2.44 | 9.96 | 11.11 |
| | Meta-UAP | -1.87 | 3.55 | 8.93 | 12.83 | 5.02 | 12.13 | 16.36 |
| | TRM-UAP | -4.79 | 0.42 | 5.37 | 5.37 | 3.33 | 10.42 | 12.40 |
| | GD-UAP | -4.54 | 0.06 | 4.61 | 7.94 | 3.18 | 10.43 | 12.18 |
| | C-PGC | 1.06 | 6.62 | 11.70 | 15.66 | 7.22 | 14.75 | 18.70 |
| | XTransfer Naive | -2.57 | 2.72 | 7.61 | 11.57 | 4.22 | 11.67 | 15.02 |
| | XTransfer Base | 9.95 | 18.5 | 24.6 | 29.3 | 17.5 | 24.2 | 33.3 |
| | **MADA-Attack(Ours)** | **27.60** | **31.12** | **33.78** | **36.35** | **34.36** | **40.93** | **44.39** |
| LLaVA-7B | AdvCLIP | 34.52 | 35.68 | 37.90 | 40.58 | -1.15 | 20.72 | 95.53 |
| | ETU | 37.45 | 39.64 | 42.99 | 46.64 | 7.36 | 27.57 | 96.06 |
| | Meta-UAP | 38.14 | 40.29 | 43.73 | 47.47 | 12.67 | 31.48 | 97.15 |
| | TRM-UAP | 37.20 | 39.35 | 42.75 | 46.56 | 3.40 | 23.61 | 95.90 |
| | GD-UAP | 36.97 | 38.98 | 42.32 | 45.86 | 0.10 | 21.79 | 95.89 |
| | C-PGC | 37.52 | 39.84 | 43.10 | 46.53 | 15.14 | 31.41 | 95.49 |
| | XTransfer Naive | 36.63 | 38.57 | 41.71 | 45.26 | 4.74 | 25.28 | 95.05 |
| | XTransfer Base | 38.79 | 41.28 | 44.73 | 48.40 | 22.25 | 38.33 | 97.05 |
| | **MADA-Attack(Ours)** | **41.86** | **44.74** | **48.24** | **51.95** | **34.34** | **47.59** | **98.25** |

and $s_{adv}$, respectively. ASR is defined as $(s_{clean} - s_{adv})/s_{clean}$. The similarity metric is task-dependent, for instance, we use CIDEr for IC, accuracy for ZS classification, and Top-K accuracy for VQA and I-T Retrieval tasks.

**Evaluation Setup** We adopt BLIP2 (Li et al., 2023), LLaVA (Liu et al., 2023a), and MiniGPT-4 (Zhu et al., 2023) as the victim models, and DINO (Oquab et al., 2023), Qwen-VL (Bai et al., 2025) and InternVL (Zhu et al., 2025) as the sophisticated ones. For ZS classification task, we use CIFAR-10 and CIFAR-100 (Krizhevsky et al., 2009), STL-10 (Coates et al., 2011), GTSRB (Stallkamp et al., 2012), Stanford Cars (Krause et al., 2013), and Food-101 (Bossard et al., 2014) for evaluation. For IC, VQA, and I-T Retrieval tasks, we use MSCOCO (Chen et al., 2015), Flickr-30K (Young et al., 2014), OK-VQA (Marino et al., 2019), and VizWiz-VQA (Gurari et al., 2018). Our surrogate models are obtained from the OpenAI releases (Radford et al., 2021), including ViT-L/14, ViT-B/16, and ViT-B/32. We apply the trained UAP to benign images to generate adversarial samples, which are then used for cross-task inference.

### 4.2. Evaluation Results

**Image Captioning Task.** Table 1 reports the non-targeted ASR on the Flickr-30K dataset across three VLMs. MADA-Attack achieves the highest ASR across all evaluated models

and captioning metrics, outperforming both single-modality and multi-modal baselines by a clear margin.

On MiniGPT-4, MADA-Attack surpasses the strongest baseline with large gains on semantic-oriented scores such as METEOR, indicating more severe semantic degradation. On the more challenging BLIP2 model, where several baselines exhibit limited effectiveness, MADA-Attack remains effective and improves CIDEr ASR from 33.3% to 44.39%, demonstrating strong robustness under difficult captioning settings. For the large-scale LLaVA-7B, MADA-Attack achieves SOTA performance on all metrics, confirming that its effectiveness scales to instruction-tuned VLM.

Compared with prior multi-modal attacks such as C-PGC, our method shows more stable performance across models, suggesting that disrupting multi-modal alignment provides a more transferable attack objective. Due to page limits, additional results on MSCOCO are reported in Appendix C, with qualitative case studies provided in Appendix D.

**Zero-shot Classification Task.** Table 2 reports the non-targeted ASR of different UAP methods on ZS classification benchmarks evaluated with BLIP2. Our method sets a new SOTA across all datasets with 82.60% accuracy, an 11.77-point improvement over XTransfer Base (70.83%). MADA-Attack shows strong transferability on challenging

*Table 2.* Non-targeted ASR (%) results of different UAP methods on zero-shot classification tasks. Higher values indicate better attack performance, and the best results are **boldfaced**.

| Method | CIFAR-10 | CIFAR-100 | Food-101 | Stanford Cars | GTSRB | STL-10 | Average |
|---|---|---|---|---|---|---|---|
| AdvCLIP | 3.07 | 7.56 | 4.03 | 5.99 | 17.08 | 0.34 | 6.34 |
| ETU | 74.68 | 89.87 | 50.33 | 32.69 | 83.41 | 17.30 | 58.05 |
| Meta-UAP | 69.53 | 87.44 | 38.05 | 18.61 | 71.72 | 11.76 | 49.52 |
| TRM-UAP | 52.75 | 68.51 | 40.30 | 21.95 | 68.92 | 7.15 | 43.26 |
| GD-UAP | 60.57 | 77.24 | 40.66 | 23.13 | 64.55 | 5.37 | 45.25 |
| C-PGC | 62.07 | 85.12 | 41.75 | 35.83 | 73.98 | 11.47 | 51.70 |
| XTransfer Naive | 79.62 | 92.54 | 60.51 | 39.67 | 84.45 | 20.56 | 62.89 |
| XTransfer Base | 86.66 | 96.44 | 72.95 | 48.95 | 90.73 | 29.27 | 70.83 |
| **MADA-Attack (Ours)** | **88.71** | **98.38** | **76.47** | **88.84** | **93.37** | **49.82** | **82.60** |

and fine-grained datasets such as Stanford Cars and STL-10 datasets, outperforming the strongest baseline by a wide margin. These results indicate that MADA-Attack generalizes effectively across diverse and distribution-shifted domains.

On standard benchmarks such as CIFAR-100 and GTSRB, MADA-Attack also achieves leading results, with ASRs of 98.38% and 93.37%, respectively, demonstrating both strong attack potency and stability. Compared with prior and multi-modal attacks such as C-PGC, MADA-Attack yields superior performance across all evaluated datasets, highlighting the effectiveness of its adversarial perturbation design in ZS classification settings. We provide evaluation results on modern models in Appendix F.

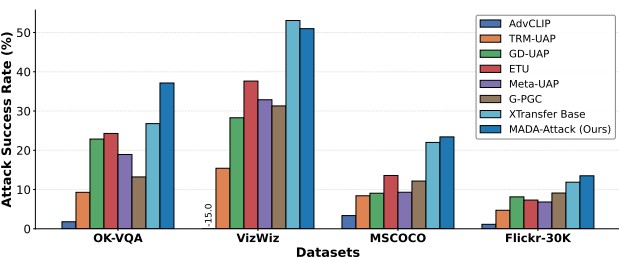

*Figure 6.* Non-targeted ASR(%) results on VQA and I-T Retrieval tasks on MiniGPT4 model. *OK-VQA* and *VizWiz* are adopted for VQA task while *MSCOCO* and *Flickr-30K* are for I-T Retrieval task, respectively. Attack results are under Recall@1.

**VQA and I-T Retrieval Tasks.** Figure 6 presents the results for the VQA and I–T Retrieval tasks evaluated on the MiniGPT-4 model. For the VQA task, MADA-Attack achieves the strongest overall performance, delivering the highest ASR on OK-VQA and remaining highly competitive on the more challenging VizWiz benchmark, while outperforming other multi-modal attack methods. These results indicate its effectiveness in disrupting knowledge-intensive and uncertainty-heavy multi-modal reasoning. For I-T Retrieval task, MADA-Attack further establishes clear advantages on both MSCOCO and Flickr-30K, surpassing XTransfer Base and all prior approaches. These results

demonstrate that MADA-Attack generalizes well across diverse vision–language tasks, highlighting its robustness and versatility in multi-modal adversarial settings.

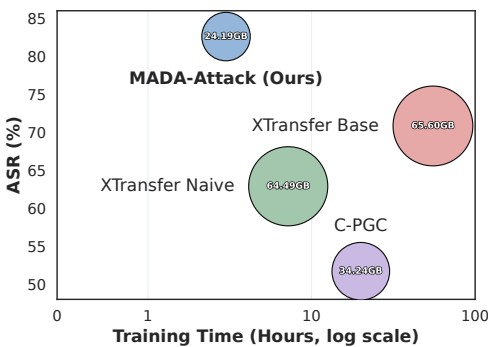

*Figure 7.* Comparison of **MADA-Attack** with other UAP methods. ASR is reported on ZS classification task. Numbers inside the markers indicate the memory consumption.

The second-best performance of MADA-Attack on VizWiz is mainly attributed to the skewed answer distribution of the dataset. A substantial portion of questions are labeled as unanswerable, and VQA models tend to default to a few dominant responses under uncertainty. Our UAP occasionally steers predictions toward these frequent answers, which overlap with the ground-truth unanswerable labels and are thus counted as correct predictions rather than attack successes. This evaluation bias leads to a partial underestimation of the actual disruption effect. Nevertheless, MADA-Attack remains competitive and outperforms other multi-modal attack methods, demonstrating its effectiveness in challenging, uncertainty-heavy VQA settings. Future improvements may further incorporate uncertain and unanswerable samples during optimization and design uncertainty-aware objectives to better capture out-of-distribution characteristics and mitigate hallucination effects.

**Training Time Analysis.** Figure 7 compares the training time of different UAP methods on ZS task. MADA-Attack achieves the highest ASR while remaining lightweight. Note

*Table 3.* Non-targeted ASR(%) across baseline attacks on close-source commercial models, and best results are **boldfaced**. Higher ASR indicates better attack performance.

| Models | Method | BLUE-1 | BLUE-2 | BLUE-3 | BLUE-4 | METEOR | ROUGE-L | CIDEr |
|--------|--------|--------|--------|--------|--------|--------|---------|-------|
| Gemini | C-PGC | 32.07 | 35.94 | 42.00 | 48.07 | 12.79 | 24.18 | 61.22 |
| | XTransfer Naive | 32.83 | 35.55 | 41.33 | 45.06 | 13.52 | 24.84 | 69.81 |
| | **MADA-Attack(Ours)** | **55.77** | **62.50** | **69.17** | **72.17** | **32.69** | **44.05** | **80.42** |
| Claude | C-PGC | 35.45 | 43.81 | 47.70 | 54.95 | 18.52 | 27.01 | 86.80 |
| | XTransfer Naive | 34.42 | 41.66 | 50.47 | 60.27 | 15.39 | 28.19 | 84.10 |
| | **MADA-Attack(Ours)** | **47.65** | **58.76** | **66.45** | **80.83** | **30.82** | **39.24** | **93.67** |
| GPT-4o | C-PGC | 35.42 | 40.42 | 46.56 | 48.17 | 13.85 | 24.44 | 70.56 |
| | XTransfer Naive | 36.52 | 43.75 | 50.55 | 52.26 | 13.44 | 26.25 | 74.50 |
| | **MADA-Attack(Ours)** | **58.30** | **65.77** | **72.22** | **76.41** | **31.81** | **45.74** | **94.59** |

that for C-PGC (Fang et al., 2025), we adopt the same hardware that our method uses, while for XTransfer Methods (Huang et al., 2025), we obtain the training times directly from the paper. Although MADA-Attack incorporates text data during training, the unified and compact text structure introduces only a marginal overhead, achieving a balance between attack strength and efficiency.

### 4.3. Evaluation on Commercial Models.

**Conventional Captioning Metrics.** Table 3 evaluates the transferability of our text-guided non-targeted adversarial attack on multiple closed-source commercial Large Vision Language Models (LVLMs), including Gemini, Claude, and GPT-4o. Despite having no access to model parameters or safety alignment mechanisms, MADA-Attack consistently achieves the best performance across all evaluation metrics, demonstrating strong black-box attack capability against proprietary multimodal systems.

Compared with existing transfer-based attack baselines, our method yields significant improvements on BLEU, METEOR, ROUGE-L, and CIDEr, indicating that the adversarial examples can effectively transfer malicious semantic intent while preserving high semantic consistency with target outputs. Notably, the performance gains remain stable across different commercial platforms with heterogeneous architectures and alignment strategies, highlighting the robustness and generalization ability of the proposed text-guided optimization framework. We provide a qualitative case studies on commercial LVLMs in Appendix G.

**Advanced Evaluation Metrics.** To further provide a more comprehensive and rigorous evaluation of attack effectiveness on commercial LVLMs, we additionally adopt the advanced LLM-as-a-judge evaluation using GPT-Scorer (Fu et al., 2024) following Li et al. (2025b). Specifically, we report Keyword Matching Rate (KMR) under three matching thresholds of 0.25, 0.5, and 1.0, denoted as $K_a$, $K_b$, and $K_c$, respectively. For the LLM-as-a-judge evaluation, we

directly use the exact evaluation prompt from the original paper to ensure fair comparison and evaluation consistency.

As shown in Table 4, MADA-Attack consistently outperforms all transfer-based baselines across all commercial LVLMs and evaluation metrics, demonstrating strong black-box transferability against proprietary multi-modal systems. Notably, the performance gains remain stable across GPT-4o, Gemini, and Claude despite their different alignment strategies and architectures, further verifying the robustness and generalization ability of our text-guided non-targeted adversarial attack. For a detailed results, see Appendix L.

*Table 4.* Partial evaluation results under $K_c$ and ASR metrics on commercial LVLMs. Higher value indicates better attack performance, and best results are **boldfaced**.

| Method | GPT-4o ($K_c$/ASR) | Gemini ($K_c$/ASR) | Claude ($K_c$/ASR) |
|--------|--------|--------|--------|
| CPGC | 0.06 / 0.02 | 0.08 / 0.03 | 0.16 / 0.11 |
| X-Transfer | 0.06 / 0.03 | 0.08 / 0.05 | 0.16 / 0.11 |
| **Ours** | **0.18 / 0.19** | **0.20 / 0.20** | **0.18 / 0.23** |

Under stricter matching criteria, MADA-Attack achieves clear improvements on $K_c$, indicating stronger semantic alignment with harmful targets rather than merely partial keyword overlap. Moreover, under the more rigorous LLM-as-a-judge evaluation setting, MADA-Attack still achieves the best ASR across all commercial LVLMs, demonstrating that the proposed attack is not limited to superficial lexical manipulation. Instead, by shifting the model's cross-modal attention toward the adversarial semantic cues embedded in the UAP, our method can more effectively alter the semantic interpretation and response generation process, leading to stronger semantic-level adversarial transfer against commercial multi-modal models.

### 4.4. Ablation Study

**Training Setup Impact.** To assess the contribution of each component, we conduct an ablation study on IC task using BLIP2 on MSCOCO dataset as shown in Table 5. We

first replace STM with random token selection. Removing STM results in a noticeable performance drop, indicating that structured textual perturbation plays a critical role in disrupting key image–text alignments. This validates our insight that jointly leveraging structured text and final-token manipulation damages semantics alignments and redirects model attention toward adversarial patterns. Next, we evaluate the impact of FET by optimizing UAP using image only. This leads to a further performance degradation beyond removing STM alone, highlighting that textual and visual optimization pathways are complementary yet modality-specific. Image-only optimization is insufficient to guide perturbations toward semantically meaningful adversarial directions, whereas incorporating textual information provides explicit semantic supervision, improving attack effectiveness.

Finally, we analyze ADA by fixing the augmentation probability. This variant suffers a moderate performance drop, corroborating our observation that early-stage optimization is crucial for shaping the global UAP structure. Aggressive augmentation disrupts semantic guidance at early iterations, causing the optimization to converge to suboptimal adversarial patterns and reducing transferability. We provide a more detailed ablation study in Appendix K.

*Table 5.* Ablation study on IC task with MSCOCO dataset and BLIP2 model, baseline indicates using the full method.

| Method | BLEU-1 | BLEU-2 | BLEU-3 | BLEU-4 | METEOR | ROUGE-L | CIDEr |
|---|---|---|---|---|---|---|---|
| Baseline | **32.46** | **38.46** | **43.09** | **47.29** | **36.46** | **40.58** | **49.42** |
| w/o STM | 26.05 | 33.13 | 39.21 | 44.42 | 30.56 | 35.22 | 45.63 |
| w/o FET | 19.98 | 28.78 | 35.86 | 41.98 | 22.89 | 27.98 | 38.99 |
| w/o ADA | 21.95 | 27.46 | 32.28 | 37.03 | 24.58 | 28.78 | 36.51 |

**Perturbation Budget Impact.** We analyze the effect of different perturbation budgets on adversarial attack performance. Specifically, we vary the image perturbation budget from $\epsilon_i = 4/255$ to the evaluation setting of $\epsilon_i = 12/255$. As a baseline, we first evaluate the benign image without perturbation, as shown in the top-left example of Figure 8. For MiniGPT-4, the generated caption describes the overall scene, identifying a living room environment, and attends to fine-grained objects such as a couch, chairs, and a table.

Under small perturbation budgets (e.g., $\epsilon_i = 4/255$ and $6/255$), the VLM begins to lose fine-grained semantic details. For instance, at $\epsilon_i = 4/255$, MiniGPT-4 fails to recognize the specific object category, describing a coffee table as a table. As the perturbation budget increases, the disruption of image–text alignment becomes more pronounced. At $\epsilon_i = 8/255$, the UAP starts to redirect the model's attention toward adversarial semantic patterns, such as the introduction of an unrelated concept (e.g., person), indicating a shift in dominant semantic focus. When the perturbation budget reaches $\epsilon_i = 12/255$, the generated caption becomes inconsistent with the original image content. As shown in the bottom-right example of Figure 8, the model produces

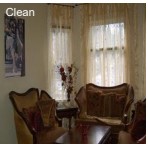 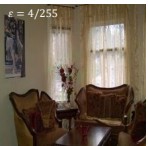 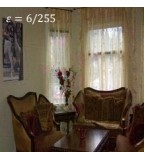

a living room with a couch, chairs, and a coffee table — a living room with a couch, chair and table — a living room with two couches and a chair

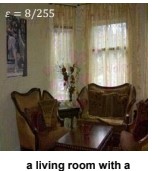 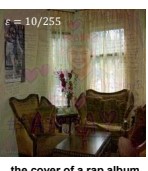 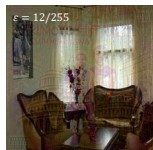

a living room with a picture of a man on the wall — the cover of a rap album with a man sitting on a couch — taxi mr goodnight mixtape

*Figure 8.* Visualization of adversarial examples with BLIP2 model with different image perturbation regulation budget.

repetitive or nonsensical text that follows the adversarial semantic cues introduced by the UAP. This behavior suggests that the original image–text alignment is overridden.

These observations support our hypothesis that UAPs follow a *disrupt-then-exploit* mechanism, where UAP first disrupt the original image–text alignment and re-align the model's attention toward adversarial semantic patterns.

## 5. Conclusion, Limitation and Future Work

In this work, we first conduct insight experiments to uncover the relationship between cross-modal attention and UAP effectiveness, revealing a disrupt-then-exploit mechanism in VLMs. Building on these observations, we propose MADA-Attack, the first framework for text-guided non-targeted UAPs in CLIP-based VLMs. By perturbing token embeddings and jointly optimizing textual and visual representations, MADA-Attack disrupts semantic alignment and achieves cross-model, cross-architecture, and cross-task transferability under black-box settings. These findings expose an underexplored multi-modal adversarial vulnerability and motivate future work to design defenses that explicitly consider multi-modal alignment and interaction patterns.

Despite the strong transferability of MADA-Attack, several limitations remain. Our study focuses on CLIP-based VLMs, and its generalization to emerging multi-modal architectures is still underexplored. In addition, the influence of data distribution and semantic diversity on UAP optimization and transferability remains insufficiently understood.

Future work will investigate more fine-grained and carefully designed multi-modal interaction mechanisms to better understand the role of cross-modal alignment in adversarial transferability. Another promising direction is to study the relationship between dataset properties and UAP optimization, including how semantic diversity and data distribution affect perturbation generalization.

## Acknowledgments

This research was supported partially by Science Challenge Project under No.TZ2025005, National Natural Science Foundation of China (62402425, 62372075, 62502432, 62302069), Natural Science Foundation of Chongqing, China (No.CSTB2024NSCQ-LZX0084), the China Postdoctoral Science Foundation under No. 2024M762829.

## Impact Statement

This work investigates UAPs in multi-modal VLMs and highlights the critical role of textual information and cross-modal alignment in shaping effective adversarial behaviors. By analyzing how textual guidance and data augmentation influence adversarial transferability, our findings provide deeper insights into the vulnerability mechanisms of modern multi-modal systems.

This study contributes to a better understanding of how VLMs integrate and rely on cross-modal semantic representations. Such insights are valuable for developing more robust and trustworthy multi-modal models, as identifying failure modes is a necessary step toward designing effective defenses, improving training strategies, and enhancing model reliability in safety-critical applications.

At the same time, we acknowledge that adversarial attack techniques may be misused to intentionally degrade the performance of deployed AI systems. We believe that the benefits of exposing and understanding these vulnerabilities outweigh the potential risks, as they enable the research community to address security concerns and strengthen multi-modal AI systems against adversarial threats. We hope that this work will encourage further research on robust multi-modal learning, responsible deployment, and principled defense mechanisms, contributing to the safe and reliable development of multi-modal artificial intelligence.

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

## A. MADA-Attack Algorithm

---

**Algorithm 1** MADA-Attack

---

**Input:** surrogate dataset $\mathcal{D}'$, surrogate CLIP models $\{\mathcal{M}_k\}_{k=1}^{K}$, optimization steps $J$, step size $\eta$, perturbation bound $\epsilon$.

**Initialize:** universal image perturbation $\delta_I$ and shared text perturbation $\delta_T$.

1: **for** $step = 1$ to $J$ **do**
2:    $(X, T) \sim \mathcal{D}'$
3:    $\tilde{X} = \mathcal{A}_t(X + \delta_I)$                      ▷ Adaptive Data Augmentation
4:    **for** each surrogate model $\mathcal{M}_k$ **do**
5:       $\tilde{h}_L^k = h_L^k + W_k(\delta_T)$              ▷ Semantic Token Manipulation
6:       $\tilde{e}_T^k = P_k(\tilde{h}_L^k), \quad \tilde{e}_I^k = f_I^k(\tilde{X})$
7:       Compute $\mathcal{L}_{text}^k(\tilde{e}_I^k, \tilde{e}_T^k)$
8:       $z^k = \mathcal{F}(e_I^k, e_T^k), \quad \tilde{z}^k = \mathcal{F}(\tilde{e}_I^k, \tilde{e}_T^k)$
9:       Compute $\mathcal{L}_{fusion}^k(z^k, \tilde{z}^k)$
10:   **end for**
11:   $\delta_T \leftarrow \delta_T + \eta_T \nabla_{\delta_T} \frac{1}{K} \sum_{k=1}^{K} \cos(\tilde{e}_I^k, \tilde{e}_T^k)$    ▷ Maximize adversarial image-text alignment
12:   $\delta_I \leftarrow \Pi_\epsilon \left( \delta_I - \eta_I \nabla_{\delta_I} \frac{1}{K} \sum_{k=1}^{K} \cos(z^k, \tilde{z}^k) \right)$   ▷ Minimize fusion similarity
13: **end for**

---

Algorithm 1 outlines the optimization procedure of MADA-Attack, which jointly learns universal perturbations in both the visual and textual modalities to disrupt multi-modal semantic alignment. At each iteration, a batch of image–text pairs is sampled from the surrogate dataset, and adversarial text is first constructed via a token-level universal text perturbation. The algorithm then computes a embedding-level distraction loss across multiple surrogate VLMs, encouraging misalignment at the uni-modal embedding stage. Subsequently, a dynamic batch-wise augmentation is applied to images, followed by the addition of a universal image perturbation, enabling the attack to account for input-level variability and improve robustness. The fusion-level loss further enforces inconsistency in the multi-modal joint representation. By averaging losses over multiple surrogate models and updating perturbations under a norm constraint, the algorithm promotes strong cross-model transferability while maintaining a single, lightweight universal perturbation.

## B. Comparison of Different Sequences of Data Augmentation

*Table 6.* Effects of Different Data Augmentation Sequences on Model Performance on MSCOCO Dataset, *Rot*, *Res*, *Flip*, *Aff* stands for rotation, resized crop, flip, affine operations, respectively.

| Method | BLEU-1 | BLEU-2 | BLEU-3 | BLEU-4 | METEOR | ROUGE-L | CIDEr |
|---|---|---|---|---|---|---|---|
| Res-Flip-Aff-Rot | **22.24** | **32.24** | **40.62** | **47.14** | **21.11** | **25.71** | **44.33** |
| Res-Rot-Flip-Aff | 17.42 | 25.59 | 32.71 | 38.79 | 17.70 | 22.28 | 35.47 |
| Rot-Aff-Res-Flip | 16.47 | 24.00 | 29.63 | 34.28 | 19.14 | 22.74 | 32.38 |
| Rot-Flip-Aff-Res | 17.50 | 27.30 | 35.61 | 42.48 | 18.28 | 22.71 | 39.62 |

Table 6 presents an ablation study on the impact of different data augmentation sequences, where the combination of Res-Flip-Aff-Rot achieves superior performance across all metrics. To interpret these results, we refer to observations from prior work on IC Attacks (Liang & Pun, 2025), which indicate that the patch embedding mechanisms inherent in modern image encoders tend to confine UAPs to localized structures. Without sufficient spatial variance, optimization converges into high-frequency *checkerboard* patterns that align with the patch grid, limiting the perturbation's transferability and adversarial potency.

We attribute the success of the Res-led sequence to the unique ability of the Resized Crop operation to disrupt this grid alignment. Unlike rotation or affine transformations which often preserve relative spatial coherence, resized cropping forces the semantic content of the original image to shift arbitrarily across the encoder's fixed patch borders during training. This spatial distinctness prevents the UAP from overfitting to specific patch locations. Instead of forming isolated artifacts within

patch boundaries, the perturbation is compelled to learn more continuous, global features that span across patch borders, breaking the checkerboard pattern.

Consequently, the Res-Flip-Aff-Rot sequence facilitates the generation of a more robust and coherent perturbation. By introducing this patch-crossing variance early in the augmentation pipeline, the optimization process is guided toward a solution that is invariant to the encoder's patch grid. This analysis is empirically supported by the significant performance gap observed in Table 4, confirming that mitigating the structural bias of patch-based processing via aggressive spatial cropping is critical for maximizing the attack performance on the MSCOCO dataset.

# C. Extended Evaluation on Image Captioning Task

*Table 7.* Non-targeted ASR(%) result in image captioning task across different VLMs on MSCOCO dataset. The best results are **boldfaced** and the second best ones are underlined, respectively.

| Model | Method | BLUE-1 | BLUE-2 | BLEU-3 | BLEU-4 | METEOR | ROUGE-L | CIDEr |
|-------|--------|--------|--------|--------|--------|--------|---------|-------|
| MiniGPT4 | AdvCLIP | 17.28 | 17.67 | 16.57 | 15.32 | 22.83 | 24.59 | 81.20 |
| | ETU | 20.90 | 22.49 | 22.30 | 21.54 | 29.48 | 28.83 | 82.89 |
| | Meta-UAP | 21.00 | 23.30 | 24.22 | 24.57 | 28.11 | 28.34 | 82.95 |
| | TRM-UAP | 19.67 | 20.77 | 20.67 | 20.42 | 27.39 | 28.00 | 82.99 |
| | GD-UAP | 21.52 | 23.08 | 23.12 | 22.85 | 33.11 | 31.84 | 83.77 |
| | C-PGC | 21.93 | 23.92 | 23.76 | 22.92 | 34.09 | 32.51 | 84.64 |
| | XTransfer Naive | 20.68 | 22.29 | 22.63 | 22.78 | 31.73 | 30.91 | 83.48 |
| | XTransfer Base | 25.39 | 29.15 | 30.91 | 31.77 | **41.14** | **37.73** | 86.22 |
| | **MADA-Attack(Ours)** | **27.89** | **31.62** | **32.97** | **33.91** | 38.34 | 36.21 | **86.64** |
| BLIP2 | AdvCLIP | 1.99 | 8.08 | 13.12 | 17.79 | 5.17 | 10.47 | 13.45 |
| | ETU | 3.24 | 7.77 | 10.34 | 12.08 | 7.67 | 9.10 | 13.69 |
| | Meta-UAP | 6.22 | 14.46 | 21.54 | 27.06 | 9.95 | 14.33 | 24.14 |
| | TRM-UAP | 3.33 | 10.45 | 16.63 | 21.65 | 7.31 | 11.51 | 18.59 |
| | GD-UAP | 2.82 | 9.49 | 15.91 | 21.27 | 7.14 | 11.53 | 18.74 |
| | C-PGC | 2.86 | 8.97 | 14.62 | 19.22 | 6.11 | 10.55 | 16.56 |
| | XTransfer Naive | 4.17 | 9.99 | 13.91 | 16.86 | 9.23 | 10.91 | 16.84 |
| | XTransfer Base | 18.54 | 29.51 | 38.13 | 44.74 | 22.83 | 26.67 | 41.92 |
| | **MADA-Attack(Ours)** | **32.46** | **38.46** | **43.09** | **47.29** | **36.46** | **40.58** | **49.42** |
| LLaVA-7B | AdvCLIP | 0.96 | 0.25 | 0.07 | -0.16 | 3.45 | 4.26 | 60.66 |
| | ETU | 6.62 | 7.56 | 10.05 | 12.45 | 14.21 | 15.93 | 65.34 |
| | Meta-UAP | 7.43 | 8.23 | 11.00 | 13.82 | 19.65 | 20.83 | 80.72 |
| | TRM-UAP | 4.92 | 5.21 | 6.94 | 8.37 | 6.62 | 6.86 | 53.77 |
| | GD-UAP | 5.33 | 6.13 | 8.71 | 10.54 | 4.27 | 6.13 | 75.93 |
| | C-PGC | 5.87 | 6.74 | 8.89 | 10.86 | 19.51 | 16.46 | 57.30 |
| | XTransfer Naive | 4.70 | 5.40 | 7.38 | 9.45 | 10.41 | 11.63 | 65.54 |
| | XTransfer Base | 9.87 | 11.22 | 14.25 | 17.29 | 30.32 | 28.85 | 77.17 |
| | **MADA-Attack(Ours)** | **17.32** | **20.02** | **23.95** | **27.29** | **44.27** | **43.43** | **84.20** |

We compare our method on MSCOCO dataset as shown in Table 7. MADA-Attack achieves the highest non-targeted ASR across all evaluated VLMs on the MSCOCO dataset, confirming its robustness on a large-scale and diverse benchmark. On MiniGPT-4, MADA-Attack reaches 27.89/31.62/32.97/33.91 on BLEU-1 to BLEU-4, surpassing the strongest baseline XTransfer Base by up to +2.50 BLEU-1 and +2.14 BLEU-4, while also achieving the best CIDEr score (86.64). The advantage becomes more pronounced on BLIP2, where our method improves BLEU-4 from 44.74 (XTransfer Base) to 47.29, and boosts CIDEr from 41.92 to 49.42, indicating stronger degradation of both syntactic structure and semantic relevance in generated captions.

Compared with existing multi-modal attacks such as C-PGC and single-modality transfer attacks such as XTransfer, MADA-Attack shows stronger cross-model stability and transferability. For example, on LLaVA-7B, MADA-Attack improves METEOR from 19.51 (C-PGC) to 44.27 and CIDEr from 57.30 to 84.20, representing substantial semantic-level gains. While XTransfer Base exhibits reasonable performance on MiniGPT-4, its effectiveness drops on more robust

architectures, whereas MADA-Attack maintains high ASR across all three VLMs. These results suggest that explicitly attacking cross-modal semantic alignment provides a more transferable and architecture-agnostic objective than visual perturbation strategies, validating the effectiveness of our proposed approach.

## D. Case Study on Image captioning task across different VLMs

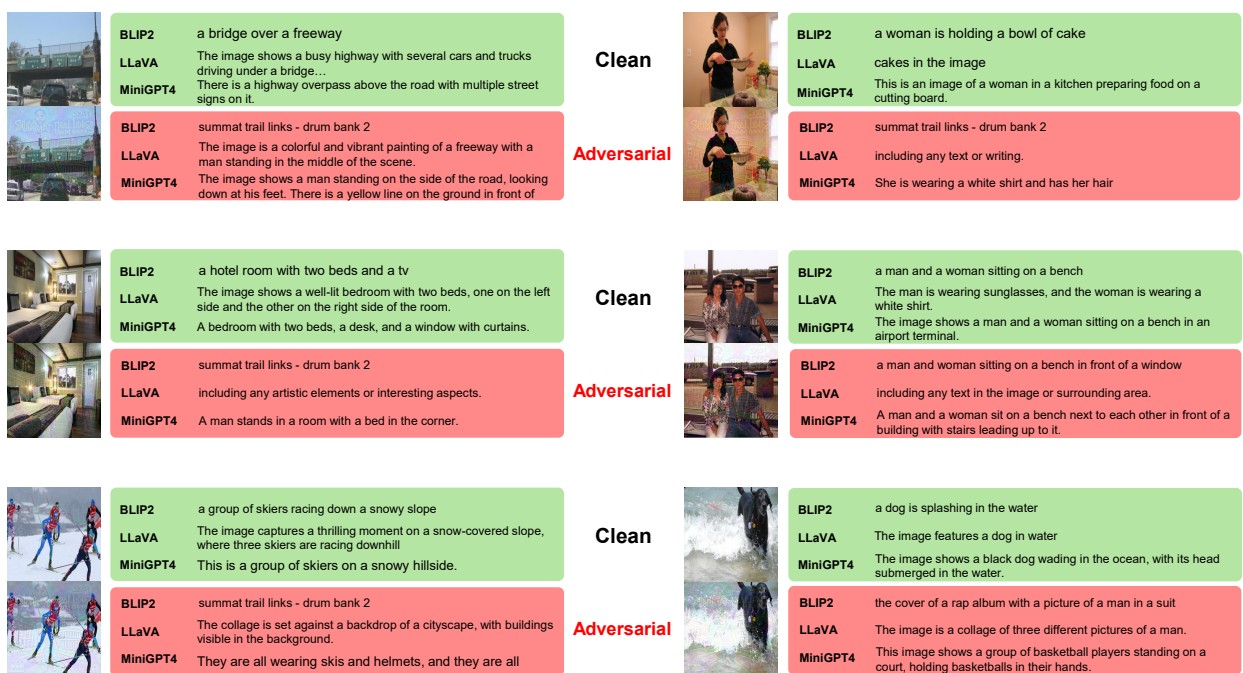

*Figure 9.* An illustration of UAP applied to an image. The top row and the bottom row represents the clean and adversarial image, respectively. Results on the MSCOCO dataset are shown in the left column, while results on the Flickr-30K dataset are shown in the right column.

Figure 9 presents case study of results on IC task on MSCOCO and Flickr-30K datasets across different VLMs. From these results, we observe two representative behaviors. In the first behavior, referred to as **Exploit**, the UAP dominates the model's visual attention, causing the model to attend to adversarial patterns rather than the original image content. This leads to captions characterized by repetitive textual structures or incorrect interpretations of global visual patterns, such as *summit trail links* or *the cover of the album*. In the second behavior, we use the term **Fused**, the UAP partially captures the model's attention while some image semantics are still retained, resulting in captions that combine adversarial content with residual visual information. Both behaviors suggest that UAP can interfere with visual–language alignment by altering the distribution of visual attention during caption generation.

## E. ADA Implementation Details

*Table 8.* Detailed parameters of ADA.

| Progress | Rotation (deg) | Resized Crop Scale | Translation Prob. | Translation Pixels |
|---|---|---|---|---|
| $p < 0.3$ | 6.0 | (0.95, 1.05) | 0.1 | {10} |
| $0.3 \leq p < 0.7$ | 10.0 | (0.9, 1.1) | 0.3 | {10, 25} |
| $p \geq 0.7$ | 15.0 | (0.8, 1.2) | 0.6 | {10, 25, 35} |

In our implementation, the augmentation policy is scheduled as a step function of training progress. Specifically, we define a piecewise-constant schedule over normalized training progress $p = \frac{\text{epoch}}{\text{total\_epochs}}$, and the augmentation sequence is fixed as: **ResizedCrop-HorizontalFlip-Affine(Translate)-Rotation**. The detailed augmentation parameters range and progressive

probability is provided in Table 8.

Motivated by prior works (Liu et al., 2025; Liang & Pun, 2025), we design a **cross-then-shift** augmentation sequence with a progressive schedule, balancing global semantic alignment and fine-grained perturbation optimization.

## F. Evaluation on Different Visual Encoders

*Table 9.* Non-targeted ASR(%) against advanced models of different architectures, and best results are **boldfaced**. Higher value represents better attack performance.

| Models | Method | CIFAR-10 | CIFAR-100 | Food-101 | GTSRB | STL-10 | Avgerage |
|---|---|---|---|---|---|---|---|
| DINOv2 | AdvCLIP | -0.01 | 0.66 | 0.03 | 1.26 | -0.04 | 0.38 |
| | ETU | 44.25 | 63.65 | 11.28 | 63.42 | 1.45 | 36.81 |
| | Meta-UAP | 66.60 | 81.40 | 11.77 | 73.78 | 5.14 | 47.74 |
| | TRM-UAP | 11.40 | 27.11 | 12.95 | 39.21 | 0.36 | 18.21 |
| | GD-UAP | 15.73 | 36.65 | 7.23 | 41.88 | 0.06 | 21.46 |
| | C-PGC | 14.22 | 63.65 | 5.17 | 42.11 | 0.15 | 25.06 |
| | XTransfer Naive | 68.08 | 82.94 | 15.36 | 67.20 | 3.27 | 47.37 |
| | **MADA-Attack(Ours)** | **87.78** | **91.31** | **18.11** | **76.50** | **5.98** | **55.94** |
| Qwen2.5-VL | AdvCLIP | -0.65 | -0.31 | 1.34 | 1.07 | 0.06 | 0.30 |
| | ETU | 70.19 | 82.84 | 48.37 | 79.02 | 15.79 | 59.24 |
| | Meta-UAP | 74.38 | 91.36 | 42.71 | 70.69 | 20.22 | 59.87 |
| | TRM-UAP | 45.76 | 65.04 | 56.61 | 73.96 | 11.66 | 50.61 |
| | GD-UAP | 48.55 | 66.25 | 39.96 | 61.69 | 6.41 | 44.57 |
| | C-PGC | 63.06 | 75.86 | 47.00 | 70.09 | 13.44 | 53.89 |
| | XTransfer Naive | 78.65 | 90.58 | **59.05** | 77.89 | 21.79 | 65.59 |
| | **MADA-Attack(Ours)** | **88.20** | **97.56** | 57.02 | **89.34** | **39.36** | **74.30** |
| InternVL3 | AdvCLIP | 0.26 | 5.49 | 1.28 | 2.45 | 0.33 | 1.96 |
| | ETU | 85.87 | 94.85 | 55.91 | 78.06 | 25.12 | 67.96 |
| | Meta-UAP | 85.06 | 95.62 | 44.15 | 78.08 | 22.67 | 65.12 |
| | TRM-UAP | 64.39 | 76.75 | 48.43 | 64.68 | 8.03 | 52.46 |
| | GD-UAP | 69.13 | 79.84 | 33.06 | 60.73 | 6.52 | 49.86 |
| | C-PGC | 74.94 | 90.31 | 44.72 | 70.04 | 14.72 | 58.95 |
| | XTransfer Naive | 84.13 | 92.49 | 47.92 | 80.42 | 16.13 | 64.22 |
| | **MADA-Attack(Ours)** | **88.00** | **96.55** | **58.17** | **85.01** | **37.29** | **73.00** |

As shown in Table 9, MADA-Attack consistently achieves the best attack performance across modern LVLMs with diverse architectures, including DINOv2, Qwen2.5-VL, and InternVL3, demonstrating strong cross-architecture transferability of the proposed text-guided non-targeted UAP. Notably, the performance gains remain stable even on advanced instruction-aligned models, indicating that our method generalizes effectively to sophisticated unseen multimodal systems.

Compared with prior transfer-based UAP attacks that mainly rely on surrogate feature alignment within CLIP-style representation spaces, MADA-Attack optimizes perturbations from the perspective of alignment distraction. By guiding the model's attention toward adversarial semantic cues that interfere with the original cross-modal alignment process, the learned perturbations are less coupled to specific backbone representations, enabling stronger semantic-level transfer across heterogeneous architectures.

## G. Analytical Case Study of Attacks on Commercial VLMs

Figure 10 presents qualitative case studies of our method on several closed-source commercial VLMs. Compared with the clean images (green boxes), which produce accurate and grounded descriptions, the adversarial images (red boxes) lead to severe semantic misalignment in the generated captions across diverse scenarios, indicating strong cross-model transferability in black-box settings. These failures are observed without access to model architectures, parameters, or training data, demonstrating that our method remains effective in realistic black-box settings against proprietary systems.

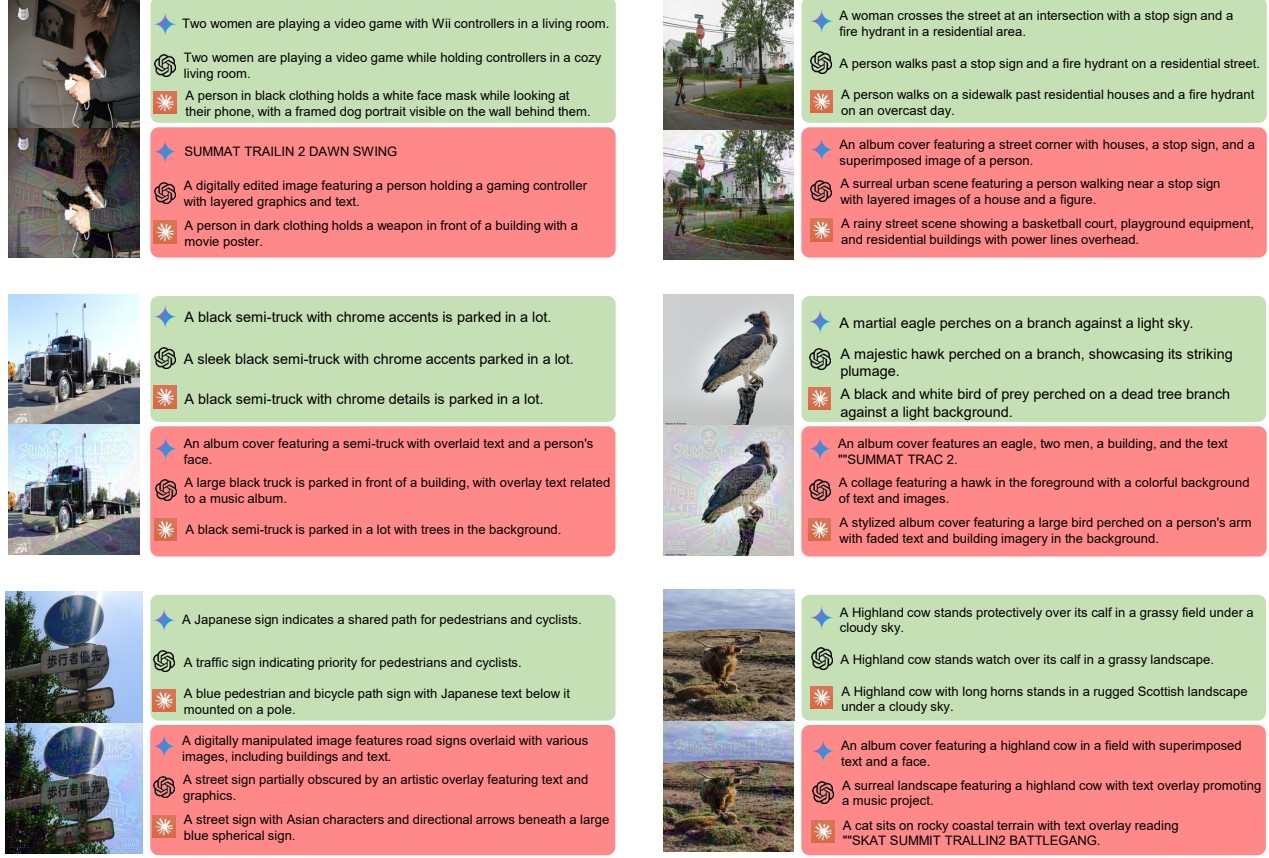

Figure 10. An illustration of UAP applied to an image across commercial models. The green box and the red box represents the clean and adversarial image, respectively.

## H. Qualitative Analysis on UAPs

Figure 11 visualizes the evolution of the UAP generated by our method across training epochs. Consistent with our insight experiments, the optimization process exhibits an observable two-stage pattern. During the early epochs, the UAP appears diffuse and unstructured, yet this phase seems to play an important role in establishing the overall optimization trajectory. As training proceeds, subsequent updates refine the perturbation into more structured, coherent patterns, suggesting a transition from coarse global guidance to more localized semantic formation.

In the final stage, the UAP features an album-cover–like style with large text-like regions, and the main body consists of salient human figures and building elements. Across a diverse set of evaluated VLMs, including BLIP2, LLaVA, MiniGPT-4, and several commercial models, these components are associated with noticeable changes in generated descriptions. In particular, text-like regions which located near the top of the UAP tend to receive increased attention, which results in repeated mentions of text or explicit statements regarding its presence. Such observations are consistent with prior findings reported in (Liu et al., 2025), which suggest that VLMs often exhibit a preference toward textual cues due to the imbalance between large language model backbones and lightweight image encoders.

We further analyze possible factors that may contribute to the emergence of such characteristic patterns. One plausible explanation relates to biases in pre-training data, where text-rich images, album covers, and architectural scenes may be associated with stronger semantic representations or higher model confidence. This hypothesis aligns with our use of category-aware unified textual guidance during UAP optimization. Another contributing factor may stem from the requirement that a single UAP generalize across heterogeneous images with varying object layouts, color distributions, and semantic focuses. In this context, large contiguous text regions and holistic cover-style designs may offer improved

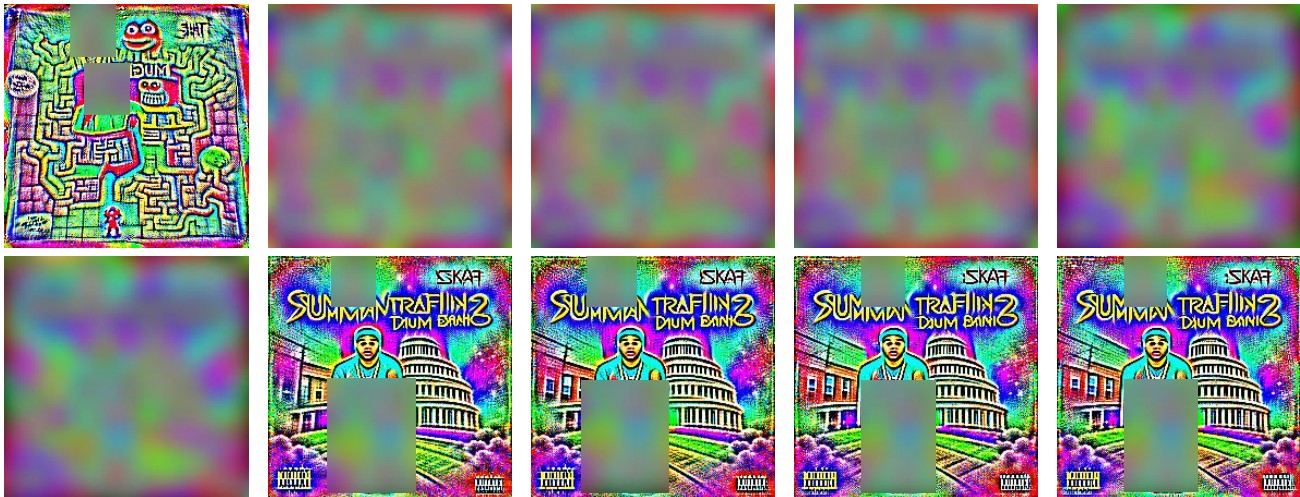

*Figure 11.* Visualization of the UAP generated by our method across training epochs. For clarity of presentation, the UAPs are amplified. Potential disturbing or inappropriate content appearing in some UAPs is blurred for ethical and readability considerations.

transferability, as they can be perceived across different visual contexts without being coupled to specific object positions.

Overall, these qualitative observations suggest that effective UAPs for VLMs tend to emphasize large-scale semantic structures, such as textual and architectural patterns, rather than fine-grained textures. Nevertheless, the mechanisms underlying UAP formation and their interaction with model representations remain insufficiently understood. We hope that this analysis provides useful empirical evidence and motivates future investigations on examining how UAP characteristics relate to pre-training data distributions and early-stage optimization dynamics, respectively.

## I. Evaluation on Adversarial Trained Encoders

*Table 10.* Non-targeted ASR(%) result in image captioning task across different VLMs on MSCOCO dataset. Higher ASR indicates better performance. The second best ones are underlined and the best results are **boldfaced**

| Victim Encoder | Method | C-10 | C-100 | Food | Cars | GTSRB | STL-10 | Average |
|---|---|---|---|---|---|---|---|---|
| TeCoA$^2$ | C-PGC | 3.30 | 5.12 | **12.35** | 5.60 | 11.87 | 1.30 | 6.59 |
| | XTransfer Naive | 2.04 | 2.59 | 8.85 | 5.10 | 6.15 | 0.53 | 4.21 |
| | XTransfer Base | 2.86 | 4.37 | 10.18 | 6.08 | 11.39 | 0.88 | 5.96 |
| | **MADA-Attack(Ours)** | **3.97** | **5.05** | 9.05 | **6.58** | **17.00** | **1.35** | **7.17** |
| Fare$^2$ | C-PGC | 7.65 | 24.02 | **15.54** | **6.42** | 18.62 | **1.75** | 12.33 |
| | XTransfer Naive | 5.52 | 16.19 | 10.93 | 5.28 | 11.55 | 1.05 | 8.42 |
| | XTransfer Base | 10.87 | 21.48 | 13.56 | 5.92 | 20.92 | 1.46 | 12.37 |
| | **MADA-Attack(Ours)** | **15.45** | **29.08** | 12.07 | 5.98 | **29.56** | 1.64 | **15.63** |

Table 10 demonstrates the results on adversarially trained encoders for the ZS classification task. We compare our approach with the SOTA UAP method XTransfer and the representative multi-modal UAP baseline C-PGC. Note that TeCoA$^2$ (Schlarmann et al., 2024) and Fare$^2$ (Mao et al., 2023) are obtained directly from the official repositories, where the superscript $^2$ indicates that the encoders are adversarially trained under a perturbation budget of $\epsilon = 2/255$.

From a comparative perspective, adversarially trained encoders are designed to defend against $L_\infty$-bounded perturbations during training. While such training improves robustness to small, localized noise, it often comes at the cost of reduced model sensitivity to fine-grained visual cues, which are crucial for ZS recognition. Consequently, the overall discriminative capability of the encoder is weakened, making it more susceptible to semantic-level disruptions introduced by universal adversarial perturbations. This explains why the performance of all evaluated UAP methods does not degrade when applied to adversarially trained encoders.

Moreover, existing UAP baselines such as XTransfer produce $L_\infty$-style perturbations, which are aligned with the threat models used during adversarial training. Under a large budget of $\epsilon = 12/255$, these perturbations are more easily suppressed by adversarially trained encoders, leading to reduced attack effectiveness. In contrast, our method targets cross-modal semantic alignment rather than pixel-level noise, a vulnerability that is not explicitly addressed by encoder-centric adversarial training, enabling more robust and consistent attack performance overall.

Finally, it is worth noting that prior work has shown that adversarial robustness can be further improved through supervised adversarial fine-tuning on large-scale datasets such as ImageNet (Mao et al., 2023), and later enhanced via unsupervised fine-tuning strategies (Schlarmann et al., 2024). While these techniques strengthen visual encoders against conventional adversarial threats, our results indicate that such improvements are still insufficient to defend against universal multi-modal attacks. This suggests that future robustness research should move beyond $L_\infty$-centric defenses and incorporate multi-modal objectives that explicitly account for cross-modal reasoning and alignment.

## J. Implementation Details

We use ImageNet (Deng et al., 2009) as the training dataset. For text construction, we adopt the classification-style template *A photo of [CLS]*. Following Zhang et al. (2024); Fang et al. (2025); Huang et al. (2025), we constrain text and image perturbations using the $L_\infty$ norm with budgets $\delta_t = 0.1$ and $\delta_i = 12/255$, respectively, and set the step size to $\eta = 0.5/255$. All experiments are conducted on 6 NVIDIA RTX 4090 GPUs. We train the model for 15 epochs with a batch size of 64. For data augmentation, we employ Kornia (Riba et al., 2020) and apply batched augmentations during training.

## K. Extend Ablation Study

We include additional ablations (-AT, -BT) of removing adversarial or benign text to better disentangle component contributions as shown in Table 11. In particular, removing either adversarial or benign text in FET leads to a dramatic performance degradation across all evaluation metrics, indicating that jointly optimizing both textual and visual branches is essential for effective universal perturbation learning. Compared with the baseline, single-sided text guidance introduces a clear modality mismatch between visual perturbation optimization and textual semantic supervision, resulting in inconsistent optimization objectives and unstable gradient directions during training. Consequently, naive text integration is not inherently beneficial and can even perform worse than conventional image-only optimization.

*Table 11.* Extend ablation study on FET module on IC task, and baseline indicates using the full method. All results are reported in non-targeted ASR(%), and higher ASR indicates better attack performance.

| Method | BLEU-1 | BLEU-2 | BLEU-3 | BLEU-4 | METEOR | ROUGE-L | CIDEr |
|---|---|---|---|---|---|---|---|
| Baseline | **32.46** | **38.46** | **43.09** | **47.29** | **36.46** | **40.58** | **49.42** |
| w/o FET | 19.98 | 28.78 | 35.86 | 41.98 | 22.89 | 27.98 | 38.99 |
| w/o AT | 6.12 | 11.30 | 16.21 | 20.84 | 7.76 | 12.63 | 15.99 |
| w/o BT | 12.97 | 20.25 | 26.96 | 32.79 | 11.97 | 16.99 | 28.18 |

To better investigate different fusion strategies in FET, we compare several representative multi-modal aggregation methods in Table 12. The results highlight the importance of carefully designed multi-modal optimization. Our method achieves the best performance across all metrics, demonstrating that the proposed min–max formulation effectively exploits cross-modal interactions for text-guided UAP optimization. In contrast, concatenation performs poorly because it treats modalities independently and fails to capture meaningful semantic interactions.

The weighted fusion results further show that increasing textual contribution improves attack effectiveness, confirming the critical role of text guidance. However, both weighted variants still underperform compared to Image-Only optimization, suggesting that naive fusion introduces gradient interference and unstable optimization due to the asymmetric objectives of text and image modalities.

*Table 12.* Comparison of different multi-modal fusion strategies on the IC task. Higher scores indicate better attack performance.

| Fusion Setting | BLEU-4 | METEOR | ROUGE-L | CIDEr |
|---|---|---|---|---|
| Image Only | 41.98 | 22.89 | 27.98 | 38.99 |
| Weighted ($\alpha = 0.7$) | 28.09 | 9.97 | 15.07 | 22.08 |
| Weighted ($\alpha = 0.3$) | 38.19 | 16.18 | 20.54 | 35.45 |
| Concatenation | 28.15 | 14.53 | 20.17 | 25.71 |
| **Ours** | **47.29** | **36.46** | **40.58** | **49.42** |

## L. Detailed Evaluation Results on Commercial LVLMs

Table 13 presents the complete GPT-Scorer evaluation results on commercial LVLMs under different KMR thresholds. Specifically, $K_a$, $K_b$, and $K_c$ correspond to progressively stricter semantic matching criteria, providing a more comprehensive evaluation of semantic consistency degradation under adversarial perturbations. The consistently strong performance across all thresholds indicates that MADA-Attack can effectively disrupt semantic alignment at multiple levels, rather than only affecting coarse textual outputs.

*Table 13.* Unified evaluation of non-targeted attacks on commercial LVLMs. Results are reported as $K_a/K_b/K_c/ASR$. Higher is better.

| Method | GPT-4o ($K_a/K_b/K_c/ASR$) | Gemini ($K_a/K_b/K_c/ASR$) | Claude ($K_a/K_b/K_c/ASR$) |
|---|---|---|---|
| CPGC | 0.24 / 0.24 / 0.06 / 0.02 | 0.22 / 0.14 / 0.08 / 0.03 | 0.30 / 0.28 / 0.16 / 0.11 |
| X-Transfer | 0.22 / 0.20 / 0.06 / 0.03 | 0.26 / 0.20 / 0.08 / 0.05 | 0.26 / 0.24 / 0.16 / 0.11 |
| **Ours** | **0.44 / 0.42 / 0.18 / 0.19** | **0.38 / 0.36 / 0.20 / 0.20** | **0.34 / 0.32 / 0.18 / 0.23** |

Compared with existing baselines, our method achieves the best overall performance across GPT-4o, Gemini, and Claude on both KMR and ASR metrics, demonstrating superior transferability under black-box settings. The consistent gains across different proprietary LVLMs further suggest that the proposed text-guided optimization strategy captures more generalizable cross-modal vulnerabilities, enabling effective attacks across diverse architectures and alignment mechanisms.

## M. Additional Analysis of FET

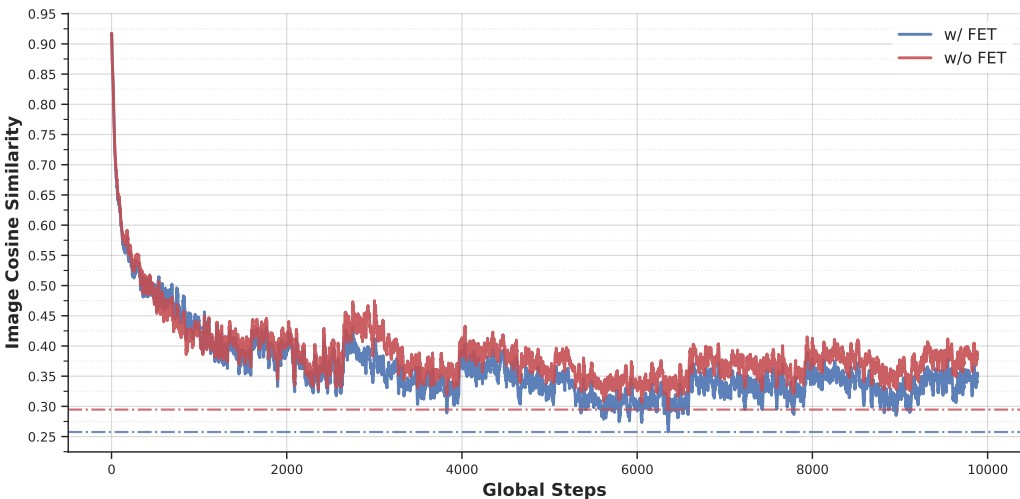

*Figure 12.* Similarity between clean and adversarial image embedding over training steps, with or without FET.

Additionally, we compare the similarity between adversarial and clean images over steps, with and without FET. The results show that FET enables faster convergence and achieves significantly stronger degradation at later stages. This further demonstrates the effectiveness of our text-guided non-targeted optimization.

# N. Comparison between Non-targeted and Targeted Attacks

While MADA-Attack demonstrates strong performance in transferable adversarial attacks, its current formulation is primarily limited to the non-targeted setting. More broadly, existing adversarial attacks against LVLMs can be divided into non-targeted and targeted paradigms, each with distinct advantages and limitations.

Non-targeted attacks emphasize universality, transferability, and scalability, aiming to induce generic misalignment behaviors across diverse prompts and models. Such methods are effective in black-box settings and often require relatively low optimization cost. However, their coarse-grained objectives usually provide limited control over the resulting adversarial behavior. In contrast, targeted attacks optimize perturbations toward specific semantic objectives, enabling fine-grained controllability and often achieving higher attack success rates. Recent studies further demonstrate that semantic relaxation and ensemble-based optimization can improve the flexibility and robustness of targeted attacks across heterogeneous LVLMs (Zhang et al., 2025b; Li et al., 2025b). Nevertheless, targeted attacks rely on stronger optimization objectives or model-specific adaptation, which may reduce scalability and transferability.

From this perspective, the difference between non-targeted and targeted attacks mainly reflects a trade-off between transferability and controllability, rather than two fundamentally separate paradigms. Both exploit vulnerabilities in multi-modal alignment and semantic representation learning. Interestingly, our observations suggest that semantic guidance may serve as a bridge between these two settings. In MADA-Attack, text-guided optimization enhances the transferability of universal perturbations while implicitly introducing a degree of semantic controllability into the attack process. This indicates the potential for unified adversarial frameworks that jointly balance transferability, efficiency, and controllability through semantic-level optimization.

# O. Theoretical Analysis of MADA-Attack

Following Qi et al., we provide a theoretical perspective on the optimization stability and transferability of MADA-Attack. Specifically, we analyze the convergence behavior of the proposed alternating optimization framework and further interpret the semantic transfer mechanism from a DRO perspective.

### O.1. Optimization Convergence

We formulate MADA-Attack as a projected min–max optimization problem:

$$\max_{\delta_I \in \Delta_I} \min_{\delta_T \in \Delta_T} \Phi(\delta_I, \delta_T),$$

where $\delta_I$ and $\delta_T$ denote image and text perturbations, respectively, and $\Delta_I, \Delta_T$ are compact constraint sets. The optimization is performed via alternating projected gradient descent (PGD) updates.

Under standard assumptions, including: (i) $L$-smoothness of $\Phi$, (ii) bounded stochastic gradient variance, and (iii) compact feasible sets, the smoothness property together with the non-expansiveness of projection operators yields:

$$\Phi(\delta_I^k, \delta_T^{k+1}) \leq \Phi(\delta_I^k, \delta_T^k) - \eta_T \|G_T^k\|^2 + \frac{L\eta_T^2}{2}\|G_T^k\|^2,$$

and

$$\Phi(\delta_I^{k+1}, \delta_T^{k+1}) \geq \Phi(\delta_I^k, \delta_T^{k+1}) + \eta_I \|G_I^k\|^2 - \frac{L\eta_I^2}{2}\|G_I^k\|^2,$$

where $G_I^k$ and $G_T^k$ denote stochastic gradients at iteration $k$, and $\eta_I, \eta_T$ are the corresponding step sizes.

By telescoping the above inequalities over $K$ iterations and choosing $\eta_I, \eta_T \leq 1/(2L)$, we obtain:

$$\frac{1}{K} \sum_{k=0}^{K-1} \mathbb{E}\left[\|G_I^k\|^2 + \|G_T^k\|^2\right] = \mathcal{O}(1/\sqrt{K}),$$

which implies that the proposed alternating optimization converges to a first-order stationary point in expectation.

## O.2. Semantic Robustness and Transferability

Beyond optimization convergence, we further analyze the transferability of MADA-Attack from the perspective of semantic robustness. Specifically, Eq. (3) can be interpreted as a distributionally robust optimization problem over semantic neighborhoods:

$$\max_{\delta_I} \mathbb{E}_x \left[ \sup_{Q_t \in \mathcal{U}_\rho(P_t)} \mathbb{E}_{t \sim Q_t} \ell_f(x, t; \delta_I) \right],$$

where $\mathcal{U}_\rho(P_t)$ denotes a semantic uncertainty set centered around the original text distribution $P_t$.

Assuming the loss function $\ell_f$ is $L_t$-Lipschitz continuous with respect to text embeddings, we have:

$$\sup_{Q_t} \mathbb{E}[\ell_f] \leq \mathbb{E}_{P_t}[\ell_f] + L_t \rho,$$

indicating that the Semantic Text Modulation (STM) mechanism approximates a worst-case semantic shift during optimization. This perspective suggests that STM improves robustness not only to individual prompts but also to semantic distribution variations.

Furthermore, for two LVLMs $f$ and $f'$ with embedding discrepancy bounded by $\|z_f - z_{f'}\| \leq \Delta_t$, the discrepancy between their DRO risks satisfies:

$$|\mathcal{R}_f^{\mathrm{DRO}} - \mathcal{R}_{f'}^{\mathrm{DRO}}| \leq C_0 + L_z \Delta_t,$$

where $L_z$ denotes the Lipschitz constant in the embedding space. This bound suggests that semantic-level optimization can reduce model-specific overfitting and improve cross-model transferability.

Finally, standard Rademacher complexity analysis provides the following empirical generalization bound:

$$\mathcal{R}_{\mathrm{gen}}(\delta_I) \leq \hat{\mathcal{R}}_{\mathrm{STM}}(\delta_I) + 2\mathfrak{R}_n(\mathcal{F}) + \sqrt{\frac{\log(1/\delta)}{2n}},$$

where $\mathfrak{R}_n(\mathcal{F})$ denotes the Rademacher complexity of the hypothesis class $\mathcal{F}$.

Overall, the above analysis suggests that MADA-Attack achieves both provable optimization convergence and improved transferability through semantic-level robustness.

