# OpenReview forum: "MADA-Attack: Transferable Multi-modal Attention Distraction Adversarial Attack against Vision Language Models"
_ICML.cc/2026/Conference — ICML 2026 regular_

### Official Review · Reviewer_UYU4 · 2026-02-20

**Soundness:** 3
**Presentation:** 3
**Significance:** 2
**Originality:** 2
**Overall Recommendation:** 4
**Confidence:** 4

**Summary:**

The paper proposes MADA-Attack, a framework for generating text-guided, non-targeted Universal Adversarial Perturbations (UAPs) against Vision Language Models (VLMs). It introduces Semantic Token Manipulation (STM) and Fused Embedding Training (FET) to disrupt cross-modal semantic alignment.

**Compliance With Llm Reviewing Policy:**

Affirmed.

**Final Justification:**

If the author could strengthen the discussion regarding advanced open-source MLLMs (such as Qwen), closed-source models (such as GPT), and image-related attacks in the final version, I believe the rating would merit an upgrade from "Weak Reject" to "Weak Accept."

**Key Questions For Authors:**

I understand that the primary scope of this paper is Universal Adversarial Perturbations (UAPs). However, the paper would be significantly stronger if the authors provided a discussion and comparison with recent image-dependent (image-specific) adversarial attacks. Even if the UAP yields a lower attack success rate than targeted, image-dependent noise, showing this comparison does not diminish the contribution of your paper. Instead, it provides the community with valuable insights into the performance gap and trade-offs between universal and image-specific perturbations. I highly recommend discussing and comparing your method against recent baselines such as:

[1] Anyattack: Towards large-scale self-supervised adversarial attacks on vision-language models, CVPR 2025.

[2] A Frustratingly Simple Yet Highly Effective Attack Baseline: Over 90% Success Rate Against the Strong Black-box Models of GPT-4.5/4o/o1, NeurIPS 2025.

**Limitations:**

The author forgot to discuss limitations.Some content from the Impact Statement could have been included.

**Strengths And Weaknesses:**

- Strengths

1. The paper addresses an important security vulnerability in multimodal systems and achieves strong empirical transferability across various tasks like zero-shot classification, image captioning, and VQA.

2. The insight regarding the "disrupt-then-exploit" mechanism in VLMs is an interesting observation

- Weaknesses

1. While the empirical results are extensive, the paper lacks the theoretical rigor. The method relies heavily on empirical observations without providing deeper mathematical or theoretical guarantees regarding the optimization dynamics or the generalization bounds of the UAPs.


2. The primary victim models evaluated in the paper (BLIP2, LLaVA-7B, and MiniGPT-4) are relatively old. To demonstrate the current relevance and true transferability of the proposed attack, the authors should evaluate it on more modern, state-of-the-art open-source models such as Qwen-VL or InternVL.

3. Table 3, which demonstrates the attack's effectiveness against closed-source commercial models (GPT-4o, Claude, Gemini), is arguably the most compelling and impactful result in the paper. However, this section is very brief, and the table is quite small. This analysis deserves significantly more space, qualitative examples, and deeper discussion in the main text.

---

> ### Author Rebuttal · Authors · 2026-03-30
>
> We greatly thank your time and careful review of our work. Below, we provide detailed responses to address each of your concerns.
>
> ---
>
> **Q1**: Theoretical Analysis
> **A1**: We thank the reviewer for the suggestion. We provide a concise theoretical analysis here and will include full proofs in the revision.
> **(1) Optimization dynamics.**
> We formulate MADA as a projected **min–max problem**:
>
> $$\max_{\delta_I \in \Delta_I} \min_{\delta_T \in \Delta_T} \;\Phi(\delta_I, \delta_T),$$
>
> optimized via alternating PGD updates. Under standard assumptions: (i) $L$-smoothness, (ii) bounded stochastic variance, and (iii) **compact** constraint sets, using smoothness and projection non-expansiveness, we obtain:
> $$
> \Phi(\delta_I^k, \delta_T^{k+1}) \le \Phi(\delta_I^k, \delta_T^k) - \eta_T \|G_T^k\|^2 + \tfrac{L\eta_T^2}{2}\|G_T^k\|^2,
> $$
> $$
> \Phi(\delta_I^{k+1}, \delta_T^{k+1}) \ge \Phi(\delta_I^k, \delta_T^{k+1}) + \eta_I \|G_I^k\|^2 - \tfrac{L\eta_I^2}{2}\|G_I^k\|^2.
> $$
> Telescoping over $K$ steps with $\eta_I,\eta_T \le 1/(2L)$ yields:
> $$
> \frac{1}{K}\sum_{k=0}^{K-1}\mathbb{E}\big[\|G_I^k\|^2 + \|G_T^k\|^2\big] = \mathcal{O}(1/\sqrt{K}),
> $$
> showing convergence to a **first-order stationary point**.
> **(2) Generalization and transferability.**
> We reinterpret Eq.(3) as a **DRO problem** over semantic neighborhoods:
> $$
> \max_{\delta_I} \mathbb E_x \Big[ \sup_{Q_t \in \mathcal U_\rho(P_t)} \mathbb E_{t \sim Q_t} \ell_f(x,t;\delta_I) \Big].
> $$
> If $\ell_f$ is $L_t$-Lipschitz w.r.t. text embeddings, then:
> $$
> \sup_{Q_t} \mathbb E[\ell_f] \le \mathbb E_{P_t}[\ell_f] + L_t \rho,
> $$
> where STM approximates this **worst-case semantic shift**.
> For cross-model transferability, if embedding discrepancies satisfy $\|z_f - z_{f'}\|\le \Delta_t$, then:
> $$
> |\mathcal R_f^{\text{DRO}} - \mathcal R_{f'}^{\text{DRO}}| \le C_0 + L_z \Delta_t,
> $$
> indicating reduced **model-specific overfitting**. Standard Rademacher bounds further ensure empirical generalization.
> **Empirical generalization.** For $n$ samples:
> $$
> \mathcal R_{\text{gen}}(\delta_I) \le \hat{\mathcal R}_{\text{STM}}(\delta_I) + 2\mathfrak R_n(\mathcal F) + \sqrt{\frac{\log(1/\delta)}{2n}}.
> $$
> Overall, MADA achieves **provable convergence** and improved **transferability via semantic robustness**. Full proofs will be provided in the revision.
>
> *[1] Lipsformer: Introducing lipschitz continuity to vision transformers, ICLR 2023.*
>
> ---
> **Q2**: Evaluation on modern models
> **A2**: Please refer to (R2, A3).
>
> **Q3**: Analysis of results on commercial models
> **A3**: We thank the reviewer for highlighting the importance of results on **closed-source commercial models**. Table below shows a subset of ASR comparisons on IC task (BLEU-4 / METEOR / CIDEr) :
>
> | Method    | Gemini          | Claude          | GPT-4o          |
> | --------- | --------------- | --------------- | --------------- |
> | C-PGC     | 48.07 / 12.79 / 61.22 | 54.95 / 18.52 / 86.80 | 48.17 / 13.85 / 70.56 |
> | XTransfer | 45.06 / 13.52 / 69.81 | 60.27 / 15.39 / 84.10 | 52.26 / 13.44 / 74.50 |
> | Ours      | **72.17 / 32.69 / 80.42** | **80.83 / 30.82 / 93.67** | **76.41 / 31.81 / 94.59** |
>
> The strong performance arises from **distribution-level text-guided optimization** in Eq. 3, which improves transferability in three aspects:
> (1) **distributional generalization**, by optimizing over textual embedding distributions rather than point-wise targets, reducing overfitting;
> (2) **semantic consistency**, as sampling $\tilde{e}_i^t$ pushes image features away from regions instead of isolated points, yielding directions aligned across models;
> (3) **robustness to model variations**, where the distributional objective produces stable perturbations under approximately Lipschitz VLM encoders.
>
> Combined with the **disrupt-then-exploit mechanism**, this enables MADA UAPs to capture **cross-model consistent vulnerabilities**, leading to strong transferability on commercial systems.
>
> ---
> **Q4**: Baseline comparison
> **A4**: We sincerely thank the reviewer for this valuable suggestion. We agree that comparisons with strong image-dependent baselines would provide a more complete picture. In the revision, we will include AnyAttack[1], Li et al. [2], and other recent methods, along with a detailed analysis.
>
> More broadly, existing attacks can be divided into **non-targeted** and **targeted** settings. Non-targeted methods emphasize **transferability and scalability**, while targeted methods achieve higher ASR with **fine-grained control**. Recent works highlight the strength of flexible, targeted attack via semantic relaxation and model ensembles.
>
> Our results suggest a potential connection between the two paradigms where **semantic guidance serve as a bridge between universal and targeted attacks**, balancing transferability, efficiency, and controllability.
>
> ---
> **Q5**: Limitations
> **A5**: We will provide a comprehensive discussion on limitation we mentioned above in revision.

---

> > ### Author Rebuttal · Reviewer_UYU4 · 2026-04-01
> >
> > I understand that the rebuttal space is limited, and for Q2–Q4 it may not be feasible to include full experimental descriptions or extensive qualitative analysis. However, some details of the experimental setup are essential for assessing the claims and for reproducibility. In particular, the response should clarify the detailed settings, e.g., whether the perturbation is only on the image modality, what perturbation budget or magnitude is used, and other implementation settings. Without these specifics, it is difficult to fully interpret the new results.
> >
> > I also want to emphasize that the results in A3 / Table 3 are among the most interesting and potentially impactful parts of the paper, especially the evaluation on commercial models. For this reason, I believe this section deserves a stronger evaluation protocol. In particular, I would recommend using more advanced metrics—for example, LLM-based judgment or scoring—to quantify ASR more reliably than surface-level matching alone. I hope the authors can address these concerns more fully in the next round of discussion.

---

> > > ### Author Response · Authors · 2026-04-02
> > >
> > > We sincerely appreciate your constructive follow-up and your emphasis on the importance of experimental clarity and rigorous evaluation. Below, we provide detailed responses to the two main points you raised.
> > >
> > > ---
> > >
> > > **(1) Clarification of experimental settings (Q2–Q4).**
> > > We agree that detailed experimental settings are important for proper interpretation and reproducibility. We provide a concise clarification here and will include full details in the final paper. Specifically:
> > >
> > > - **Perturbation modality:** The perturbation is applied only to the **image modality**.
> > > - **Perturbation budget:** All attacks is under $L_\infty$ constrains with **perturbation budget** $\epsilon=12/255$ following prior work [1],[2],[3].
> > > - **Other implementation details:** We provide an implement details in **Appendix H**.
> > >
> > > These details will be explicitly added to evaluations on LVLMs to improve clarity and reproducibility. Our code will be released upon acceptance.
> > >
> > > [1] One Perturbation is Enough: On Generating Universal Adversarial  Perturbations against Vision-Language Pre-training Models, ICCV 2025.
> > > [2] Universal adversarial perturbations for vision-language pre-trained models, ACM SIGIR 2024.
> > > [3] X-Transfer Attacks: Towards Super Transferable Adversarial Attacks on CLIP, ICML 2025.
> > >
> > > ---
> > >
> > > **(2) Strengthened evaluation on commercial models.**
> > > We thank the reviewer for highlighting the importance of strengthening the evaluation protocol, particularly for commercial models. We agree that the results in A3 / Table 3 are critical and require more rigorous and reliable assessment.
> > > **(a) Metrics.**
> > > Regarding the reviewer’s suggestion, we adopt an **LLM-as-a-judge** paradigm. We follow M-Attack [1] and report **LLM-based evaluation metrics**:
> > >
> > > - **Attack Success Rate (ASR)**
> > > - **Keyword Matching Rate (KMR)**
> > >
> > > Specifically, we employ ***GPTScore*** [2] as the evaluator and use the **same prompt as in the original work** to ensure fairness and reproducibility.-
> > >
> > > **(b) Adaptation.**
> > > However, these metrics are originally designed for **targeted attacks**, where similarity is calculated between **target and adversarial text**. To better adapt to the **non-targeted settings**, we instead measure the **relative similarity drop** between clean and adversarial outputs:
> > > $$
> > > ASR = \frac{\mathrm{sim}(\text{clean}, \text{ref}) - \mathrm{sim}(\text{adv}, \text{ref})}{\mathrm{sim}(\text{clean}, \text{ref})}
> > > $$
> > >
> > > $$
> > > KMRScore = \frac{\mathrm{KMR}(\text{clean}, \text{ref}) - \mathrm{KMR}(\text{adv}, \text{ref})}{\mathrm{KMR}(\text{clean}, \text{ref})}
> > > $$
> > >
> > > The results are summarized below. Following [1], we report KMR under three thresholds (0.25, 0.5, 1.0), denoted as $K_a$, $K_b$, and $K_c$, respectively.
> > >
> > > **Table 11. Unified evaluation of non-targeted attacks on commercial LVLMs (higher is better).**
> > >
> > > | Method                 | GPT-4o ($K_a$ / $K_b$ / $K_c$ / ASR) | Gemini ($K_a$ / $K_b$ / $K_c$ / ASR) | Claude ($K_a$ / $K_b$ / $K_c$ / ASR) |
> > > | ---------------------- | ------------------------------------ | ------------------------------------ | ------------------------------------ |
> > > | CPGC                   | 0.24 / 0.24 / 0.06 / 0.02            | 0.22 / 0.14 / 0.08 / 0.03            | 0.30 / 0.28 / 0.16 / 0.11            |
> > > | X-Transfer             | 0.22 / 0.20 / 0.06 / 0.03            | 0.26 / 0.20 / 0.08 / 0.05            | 0.26 / 0.24 / 0.16 / 0.11            |
> > > | **MADA-Attack (Ours)** | **0.44 / 0.42 / 0.18 / 0.19**        | **0.38 / 0.36 / 0.20 / 0.20**        | **0.34 / 0.32 / 0.18 / 0.23**        |
> > >
> > > **(c) Results and analysis.**
> > > As shown in Table 11, *MADA-Attack* outperforms all baselines across all models and metrics, demonstrating strong transferability to commercial LVLMs. In particular, MADA achieves substantially higher ASR values (e.g., 0.19 vs. 0.02–0.03 on GPT-4o), indicating significantly stronger **semantic degradation**. Meanwhile, the improvements are consistent across all KMR thresholds, suggesting both **broader attack coverage** and more **severe perturbation effects**. Notably, these gains remain stable across different LVLMs (GPT-4o, Gemini, and Claude), highlighting the robustness and generalization ability of the proposed method.
> > >
> > > Overall, the results are consistent with those observed under traditional metrics, further validating the effectiveness of *MADA-Attack* from both **lexical-level** and **semantic-level** perspectives. These additional evaluations complement conventional metrics by capturing semantic-level degradation more effectively, thereby providing a more comprehensive and reliable assessment of attack performance. We will incorporate these results and corresponding analysis into the revision in analysis on commercial models to further strengthen the empirical study.
> > >
> > > [1] A Frustratingly Simple Yet Highly Effective Attack Baseline: Over 90% Success Rate Against the Strong Black-box Models of GPT-4.5/4o/o1, NeurIPS 2025.
> > > [2] GPTScore: Evaluate as You Desire, ACL 2024

---

### Official Review · Reviewer_kYb4 · 2026-03-06

**Soundness:** 2
**Presentation:** 3
**Significance:** 3
**Originality:** 2
**Overall Recommendation:** 4
**Confidence:** 4

**Summary:**

This paper proposes MADA-Attack, a transferable universal adversarial perturbation method for vision-language models. The approach jointly optimizes visual and textual embeddings to disrupt cross-modal alignment, using semantic token manipulation and adaptive training strategies. Experiments across multiple tasks show improved attack transferability compared to prior UAP methods.

**Compliance With Llm Reviewing Policy:**

Affirmed.

**Final Justification:**

While the technical novelty remains incremental relative to prior work, the rebuttal has adequately addressed my initial concerns. In light of these clarifications, I have decided to upgrade my overall score from 3 to 4.

**Key Questions For Authors:**

Please refer to the points raised in the **Weaknesses** section.

**Limitations:**

yes

**Strengths And Weaknesses:**

## Strengths

1. The paper is clearly written and easy to follow. The experimental section is particularly well organized.

2. The paper proposes a multi-modal joint optimization framework, **MADA-Attack**, which introduces textual perturbations into the generation of universal adversarial perturbations (UAPs) and jointly optimizes visual and textual embeddings.

3. The experiments cover multiple tasks and models, and the results demonstrate strong attack effectiveness.

---

## Weaknesses

1. The paper claims to be the first to study text-guided non-targeted UAPs for VLMs. However, the core components (STM, FET, ADA) appear to be incremental combinations of ideas already present in **C-PGC, Co-Attack, MSI-Attack**, as well as standard adaptive augmentation techniques.

2. MADA-Attack is heavily built upon **CLIP-based surrogate models** and CLIP-style cross-modal embeddings. It remains unclear to what extent the method transfers to modern non-CLIP VLMs (e.g., **Qwen2.5-VL** and **InternVL3** that use custom vision encoders). If transferability is limited, this would significantly reduce the claimed generality of the attack.

3. In **FET**, given that visual tokens dominate shallow layers while textual ones govern deeper layers, why does Eq. 4 optimize textual perturbation $\delta^T$ using benign image embeddings $f_I(x_i)$ instead of adversarial ones $f_I(x_i + \delta^I)$? Would adversarial embeddings better coordinate the layer-wise dynamics, and are there ablations comparing the two?

4. The ablation experiments (e.g., Table 4) claim to validate **FET** as the core for joint visual-textual optimization, but they are not clean or comprehensive. Specifically, the **"w/o FET"** variant simply reverts to image-only optimization, which fails to isolate and quantify the individual contributions of FET's sub-components, such as the additive fusion operator $\mathcal{F}(\cdot)$ (why not alternatives like concatenation, weighted sum, or cosine coupling?) or the relative impacts of the **Interaction Phase vs. Fusion Phase**. This coarse design leaves key questions unanswered and may overstate FET's necessity.

5. While **FET** emphasizes a two-phase approach (first optimizing textual perturbation $\delta^T$ in Eq. 4, then fusing into visual perturbation $\delta^I$ in Eq. 5) for coordinated misalignment, the experiments do not demonstrate its superiority over **simultaneous updates** (e.g., joint optimization of $\delta^T$ and $\delta^I$ in a single phase). Ablations like **"w/o FET"** do not address this issue. Without targeted comparisons (e.g., ASR drops under simultaneous vs. sequential optimization), the design choice appears arbitrary and weakens the claims regarding efficiency and transferability.

---

> ### Author Rebuttal · Authors · 2026-03-30
>
> We greatly thank your thoughtful review and valuable suggestions. Below we address each of your raised concerns in detail.
>
> ---
> **Q1**: Novelty of our work
> **A1**: **(1) Prior work differences & motivation.**
> Prior multimodal attacks mainly focus on **pair-wise I–T perturbations** for specific tasks (e.g., retrieval), where image and text are optimized **cooperatively** to maximize embedding discrepancy per input. Such designs are tightly coupled to the evaluation setting and do not explicitly address **transferability**.
>
> **(2) Intuition & framework novelty.**
> We instead study **transferable text-guided** non-targeted UAPs across models, tasks, and datasets. We formulate a **min–max objective** where text perturbations act as a **worst-case semantic anchor**, shifting the goal from pair-wise discrepancy to **task-agnostic transferability**.
> Based on this insight, our method is not an incremental combination of modules, but a **principled, unified design** where $\delta^T$ explicitly guides the optimization of $\delta^I$:
> - **STM**: provides unified semantic guidance,
> - **FET**: enables modality-consistent interaction,
> - **ADA**: stabilizes optimization to better exploit text signals.
> These components are jointly motivated by the min–max formulation, rather than independently added. They enable a **disrupt-then-exploit** mechanism for transferable attacks. We further analyze modality differences in VLMs and demonstrate strong **cross-task and cross-model transferability**, achieving **SOTA performance**.
>
> ---
> **Q2**: Transferability beyond CLIP-based models
> **A2**: Due to space limits, please refer to (Reviewer nsnL, A3).
>
> **Q3**: Alternatives in Eq. 4
> **A3**: We thank the reviewer for the recognition of our insights , and we acknowledge that our original formulation was indeed incorrect. Specifically, the image embedding in Eq. 4 should be based on the **adversarial image** rather than the benign input. The correct formulation is:
>
> $\delta^T_{j+1}=\delta^T_j-\eta\nabla_{\delta^T}\mathrm{dist}(f_I(x_i+\delta^I),f_T(t_i)+\delta^T_j)$
>
> The corrected formulation enforces that the outer maximization finds an image perturbation that induces a **worst-case semantic shift**, i.e., a discrepancy that persists even after optimal alignment, clarifying the intended use of adversarial versus benign embeddings. We will carefully revise the presentation to ensure the formulation is precise.
>
> ---
> **Q4**: Coarse ablation
> **A4**: **(1) Extended ablations.** We include additional ablations (-AT, -BT) of removing adversarial or benign text to better disentangle component contributions as shown below.
>
> | Method | METEOR | CIDEr |
> | ------ | ------ | ----- |
> | -FET   | 22.89  | 38.99 |
> | -AT    | 7.76   | 15.99 |
> | -BT    | 11.97  | 28.18 |
>
> The ablation results show that introducing text on only one side causes **modality mismatch**, leading to misaligned objectives and unstable gradients, and indicating that naive text integration is **not inherently beneficial**, and even worse than image-only optimization.
> Such result underscores the importance of FET for modality-consistent optimization in effective UAP updates. We will include more comprehensive ablations in the revision.
>
> **(2) Fusion operator.** Our goal is to enable minimal yet effective cross-modal interaction rather than introducing complex modules. Prior work [1] shows that increasing text contribution improves attack effectiveness, supporting our insight that text provides strong adversarial guidance. Our design realizes this benefit with **minimal overhead**, while more sophisticated fusion strategies are orthogonal to our design and could be explored in future work.
>
> **(3) Roles of phases.** The two phases serve **complementary roles**:
> - Interaction Phase identifies a **worst-case semantic anchor**, capturing the semantic boundary/direction;
> - Fusion Phase translates this into a concrete **optimization objective** for updating the image perturbation.
> As our response in A1, both phases are necessary for text-guided UAPs.
> ---
> **Q5**: Sequential vs. Joint Optimization
> **A5**: The sequential optimization solves the min–max problem in Eq. 3 by enforcing text-guided UAP learning, thereby enhancing transferability through disrupting the alignment mechanism in VLMs. In contrast, simultaneous updates of $\delta_T$ and $\delta_I$ introduce two key issues:
> - **gradient conflicts** from optimizing heterogeneous embeddings, leading to unstable update directions and suboptimal perturbations;
> -  the **lack** of prior **semantic guidance**, where $\delta_I$ may follow directions that do not effectively increase misalignment.
>
> Together, these issues introduce irrelevant signals and conflicting objectives, resulting in unstable optimization dynamics and convergence to suboptimal flat regions.
> *[1] Chain of Attack: On the Robustness of Vision-Language Models Against Transfer-Based Adversarial Attacks, CVPR 2025*

---

> > ### Author Rebuttal · Reviewer_kYb4 · 2026-04-02
> >
> > Thank you for the rebuttal and the additional experiments. While some of my concerns have been addressed, the following issues remain:
> >
> > (1) The core idea of the paper is relatively straightforward and offers only limited novelty. The notions of double optimization problem, fused embeddings, and adaptive data augmentation are well-established in the existing literature and do not appear to introduce meaningful new insights.
> >
> > (2) The paper lacks a clear description of the specific fusion operation employed (e.g., whether it is a simple concatenation of features). Furthermore, despite my earlier concern, no ablation study was performed to examine the contribution or design choices of this fusion mechanism.

---

> > > ### Author Response · Authors · 2026-04-04
> > >
> > > We thank the reviewer for the thoughtful follow-up. We clarify that our contribution is not a simple combination of existing components, but is grounded in key underlying insights that guide a progressively developed design. Below we provide detailed responses to these concerns.
> > >
> > > ---
> > > **Q1: Novelty justification.**
> > >
> > > **(i) Paradigm difference.**
> > > Prior multi-modal attacks typically follow a **co-optimization paradigm**, treating image and text as **independent attack sources** with **instance-level objectives** (e.g., generating adversarial I-T pairs via separate perturbations). In such settings, text is weakly coupled with image optimization and often does **not substantially transfer** to image-only attacks.
> > > In contrast, we study **text-guided UAP optimization**, where text serves as a **guidance signal** to improve **distribution-level transferability** of image perturbations.
> > >
> > > **(ii) New challenges.**
> > > Adapting prior methods to this setting introduces two key challenges that cannot be resolved by naively combining off-the-shelf components:
> > > - **Objective unification:** Ensuring text acts as guidance rather than an independent attack path;
> > > - **Cross-modal alignment:** Transferring textual signals into the image representation space.
> > >
> > > **(iii) Insight-driven design.**
> > > Our three key components addresses these challenges through three key insights:
> > > - **(a) Mechanism → STM.** We identify a _“disrupt-then-exploit”_ mechanism in UAP under multi-modal alignment and design **STM** to transform instance-level text into **unified guidance** for distribution-level optimization.
> > > - **(b) Interaction → FET.** We analyze modality roles in VLMs and propose **FET** to **bridge cross-modal interactions**, enabling consistent text-guided optimization in the image space.
> > > - **(c) Trade-off → ADA.** We observe a **guidance–diversity trade-off** and introduce **ADA** to stabilize optimization while preserving effective guidance.
> > >
> > > **(iv) Qualitative improvement, not incremental combination.**
> > > While fusion and augmentation are common, naively introducing text or directly applying multi-modal techniques can even degrade performance in text-guided non-targeted attack settings, revealing **inherent cross-modal conflicts**. In contrast, Our **text-to-image pipeline** resolves these issues by leveraging an adversarial **min–max design** that treats both modalities **equally** and enables **minimal yet effective** cross-modal interaction to **unify the objectives**, resulting in a **qualitative improvement in transferability**, rather than incremental gains.
> > >
> > > We will further clarify these distinctions in the introduction section.
> > >
> > > ---
> > > **Q2: Fusion ablation.**
> > > We thank the reviewer for pointing out the lack of clarity regarding the fusion operation and the
> > > absence of ablation analysis. We further conduct **comprehensive ablation studies** on fusion actions, summarized below.
> > >
> > > | Fusion Setting   | BLEU-4    | METEOR    | ROUGE-L   | CIDEr     |
> > > | ---------------- | --------- | --------- | --------- | --------- |
> > > | Image Only       | 41.98     | 22.89     | 27.98     | 38.99     |
> > > | Weighted (α=0.7) | 28.09     | 9.97      | 15.07     | 22.08     |
> > > | Weighted (α=0.3) | 38.19     | 16.18     | 20.54     | 35.45     |
> > > | Concatenation    | 28.15     | 14.53     | 20.17     | 25.71     |
> > > | **Ours**         | **47.29** | **36.46** | **40.58** | **49.42** |
> > > **(i) Key observations.**
> > > - **Effective adversarial bilevel optimization.** Our method achieves the best performance across all metrics, showing that the proposed **min–max formulation** enables effective **text-guided optimization**. Treating image and text modalities **symmetrically** allows better exploitation of cross-modal signals for UAP optimization.
> > > - **Concatenation lacks cross-modal interaction.** Concatenation underperforms as it treats modalities independently and fails to capture **cross-modal interactions**, leading to suboptimal alignment and weaker transferability.
> > > - **Text-Guided Enhancement.** Weighted fusion shows that increasing text contribution (smaller α) improves attack strength, highlighting the **critical role of text guidance** in shaping the optimization landscape.
> > > - **Weighted sum introduce gradient interference.** Weighted fusion still underperforms *Image-Only* due to imbalanced coupling. Since text and image are optimized in opposite directions (min vs. max), **biased fusion** introduces **misaligned gradients** and **unstable optimization**.
> > >
> > > **(ii) Additional evidence.**
> > > Additionally, we compare the similarity between adversarial and clean images over steps, with and without FET. The results show that FET enables **faster convergence** and achieves significantly **stronger degradation** at later stages. This further demonstrates the effectiveness of our **text-guided non-targeted** optimization. The corresponding visualization is available at the anonymous [link](https://anonymous.4open.science/r/Sim-over-steps/).

---

### Official Review · Reviewer_nsnL · 2026-03-12

**Soundness:** 3
**Presentation:** 3
**Significance:** 3
**Originality:** 3
**Overall Recommendation:** 4
**Confidence:** 3

**Summary:**

This paper studies adversarial robustness in Vision-Language Models (VLMs) and proposes a framework for generating universal adversarial perturbations (UAPs) in multimodal systems called MADA-Attack.

The authors argue that many existing UAP approaches focus mainly on the visual modality and neglect cross-modal semantic alignment between images and text. To address this issue, the paper analyzes cross-modal attention behavior in CLIP-based models and introduces a multimodal attack framework consisting of three components: Semantic Token Manipulation (STM), Fused Embedding Training (FET), and Adaptive Data Augmentation (ADA).

Experiments across multiple tasks—including zero-shot classification, image captioning, VQA, and image–text retrieval—show that the proposed method improves attack transferability and achieves strong empirical performance across several VLM architectures.

**Compliance With Llm Reviewing Policy:**

Affirmed.

**Final Justification:**

My concerns have been addressed, so I am maintaining my positive score.

**Key Questions For Authors:**

**Q1. ADA scheduling details**

Could the authors provide the exact functional form of the augmentation scheduling strategy used in ADA?
For example, it would be helpful to include the augmentation probability curve with respect to training iterations (e.g., how the probability evolves during training) and a sensitivity analysis of different schedules.

---

**Q2. Role of textual perturbation during inference**

From the description of the optimization procedure, it appears that the textual perturbation $\delta_T$ is introduced primarily during the **training stage** to guide the optimization of the image UAP. My understanding is that the final attack at inference time only applies the learned image perturbation $\delta_I$, without modifying the textual input.

If this interpretation is correct, it may be helpful for the paper to explicitly state that $\delta_T$ is used only during training and is not required during inference.

---

**Q3. Transferability across different encoder architectures**

Have the authors evaluated MADA-Attack on VLMs using substantially different visual encoders or multimodal architectures? For example, testing on models based on encoders such as DINOv2, ConvNeXt, or SigLIP could help verify whether the observed transferability generalizes beyond CLIP-style backbones.

**Limitations:**

yes

**Strengths And Weaknesses:**

### Strengths

**S1. Clear motivation and insightful analysis**

The paper provides a clear motivation by analyzing cross-modal attention behavior in VLMs and identifying limitations of image-only UAP attacks. The proposed “disrupt-then-exploit” interpretation offers an intuitive explanation of how adversarial perturbations affect multimodal alignment.

**S2. Multimodal attack framework**

The method integrates textual guidance into UAP optimization through STM and FET, forming a coherent multimodal attack pipeline. The idea of optimizing perturbations in a semantic embedding space is an interesting design choice.

**S3. Broad experimental coverage**

The paper evaluates the proposed approach across multiple tasks and models, including classification, captioning, VQA, and retrieval. The experiments show consistent improvements over several baseline methods.

---

### Weaknesses

**W1. Transferability evaluation may be influenced by shared visual backbones**

Several evaluated VLMs appear to rely on CLIP-style visual encoders or closely related architectures. Therefore, part of the observed transferability may come from shared backbone representations rather than the proposed multimodal attack mechanism itself. Additional experiments on models using different visual encoders (e.g., DINOv2, ConvNeXt, or SigLIP) would strengthen the claim of cross-architecture transferability.

---

**W2. Limited details of the adaptive augmentation schedule**

Although the Adaptive Data Augmentation (ADA) module appears to play an important role in improving transferability, the description of the dynamic scheduling mechanism remains somewhat vague. The paper states that the augmentation probability increases during training but does not clearly specify the scheduling function or hyperparameters.

Providing additional details—such as the augmentation probability curve with respect to training iterations, the initial and final probabilities, or a sensitivity analysis of different schedules—would help other researchers better reproduce the proposed mechanism.

---

> ### Author Rebuttal · Authors · 2026-03-30
>
> We sincerely appreciate your insightful comments and useful suggestions. We provide detailed responses to each of your concerns below.
>
> ---
> **Q1**: ADA scheduling details
> **A1**: **(1) Augmentation Details**. In our implementation, the augmentation policy is scheduled as a **step function** of training progress. Specifically, we define a **piecewise-constant** schedule over normalized training progress $p = \frac{\mathrm{epoch}}{\mathrm{total\_epochs}}$:
>
> | Progress | Rotation (deg) | Resized Crop Scale | Translation Prob. | Translation Pixels |
> |:--------:|:--------------:|:------------------:|:-----------------:|:------------------:|
> | p<0.3    | 6.0            | (0.95, 1.05)       | 0.1               | {10}               |
> | 0.3≤p<0.7 | 10.0         | (0.9, 1.1)         | 0.3               | {10, 25}           |
> | p≥0.7   | 15.0           | (0.8, 1.2)         | 0.6               | {10, 25, 35}       |
>
> The augmentation sequence is fixed as: **ResizedCrop-HorizontalFlip-Affine(Translate)-Rotation**.
>
> **(2) Intuition.** Prior work identifies two key factors limiting transferability in transformation-based attacks:
> - Patch-wise processing in ViTs induces locality artifacts, which can be mitigated by **cross-pixel interactions** [1];
> -  **Trade-off** exists between data diversity and semantic preservation, where improper augmentation strength degrades performance [2].
>
> **Motivated by these insights**, we design a **cross-then-shift augmentation sequence** with a progressive schedule, balancing global semantic alignment and fine-grained perturbation optimization.
> - **Early-stage** mild augmentations preserve **semantic consistency**, enabling stable UAP optimization within the original image-text alignment space.
> - **Later-stage** stronger transformations introduce **cross-pixel diversity**, encouraging the perturbation to learn more transferable, fine-grained structures (as visualized in Fig. 11).
>
> **(3) Sensitivity**. Importantly, our design relies on **minimal yet structured transformations**, rather than aggressive tuning. As shown in Appendix B (Table 5), the proposed sequence consistently outperforms alternatives, while varying sequences yields performance within a relatively narrow range (16.47%–17.50%), indicating that the method is **not overly sensitive** to specific schedules and is reproducible in practice. We will incorporate this clarification into the revised version to better motivate ADA and its robustness.
>
> *[1] IC Attack: In-place and Cross-pixel Augmentations for Highly Transferable Transformation-based Attacks. ACM MM 2025.*
> *[2] Learning to transform dynamically for better adversarial transferability, CVPR 2024.*
>
> ---
> **Q2**: Role of textual perturbation during inference
> **A2**: We thank the reviewer for the careful reading. The understanding is correct: the textual perturbation is **only used during training** to guide the optimization of the image UAP, and **is NOT required at inference time**. At test time, the attack is performed by directly applying the learned image perturbation, without modifying the textual input.
>
> Such design ensures that the benefit comes from **improved perturbation learning**, rather than requiring additional modalities at inference, making the attack more **practical** and broadly **applicable**. We will clarify this explicitly in the revised version.
>
> ---
> **Q3**: Transferability across different encoders
> **A3**: **(1) Extend Evaluation**. We agree that evaluating on models with substantially different visual encoders is important to better verify the robustness and generalization of *MADA-Attack*. To this end, we further evaluate attacks on modern VLMs of **diverse architectures** across **5 datasets**. Due to space limitations, we report a subset of representative results, and for a detailed one, please refer to [Anonymous Link](https://anonymous.4open.science/r/ICML-Rebuttal-16610-result-on-different-models/Table%208.pdf).
>
> | Model      | XTransfer | Ours      |
> | ---------- | --------- | --------- |
> | DINOv2     | 47.37     | **55.94** |
> | Qwen2.5-VL | 65.59     | **74.30** |
> | InternVL3  | 64.22     | **73.00** |
>
> The results demonstrate that *MADA-Attack* maintains consistent **SOTA performance** across different encoder designs, further supporting its robustness and generalization ability. We will include these additional experiments in the revised version to provide a more comprehensive evaluation. Additionally, Table 5 demonstrate MADA-Attack successfully disrupt commercial model's interpretation, confirming the generalization and robustness for adversarial transferability.
> **(2) Reasons for using CLIP-style encoders.** We use CLIP-based models as they are a **mainstream benchmark** in transferable attacks due to strong cross-modal alignment and fair comparability. Strong performance on architecturally different closed-source systems (Table 5) further demonstrates transferability beyond CLIP-style backbones.

---

> > ### Author Rebuttal · Reviewer_nsnL · 2026-04-03
> >
> > My concerns have been addressed, so I am maintaining my positive score.

---

> > > ### Author Response · Authors · 2026-04-04
> > >
> > > We sincerely thank the reviewer for the thoughtful and constructive feedback. We particularly appreciate the questions regarding ADA scheduling, the role of textual perturbations during inference, and transferability across architectures. These comments have helped us significantly improve the clarity and completeness of the paper. We have carefully incorporated these points into our revision and believe they substantially enhance both the clarity and rigor of the work.

---

### Official Review · Reviewer_MFYv · 2026-03-13

**Soundness:** 3
**Presentation:** 3
**Significance:** 3
**Originality:** 2
**Overall Recommendation:** 4
**Confidence:** 3

**Summary:**

This paper proposes MADA-Attack, a novel framework for generating transferable universal adversarial perturbations (UAPs) against vision-language models (VLMs). The method addresses key limitations of existing approaches by jointly optimizing perturbations in both visual and textual modalities through three core components: Semantic Token Manipulation (STM) for disrupting text-guided attention patterns, Fused Embedding Training (FET) for coordinated cross-modal misalignment, and Adaptive Data Augmentation (ADA) for balancing attack strength and generalization. Extensive experiments demonstrate that MADA-Attack achieves state-of-the-art performance across multiple tasks including zero-shot classification, image captioning, VQA, and image-text retrieval, with an average attack success rate (ASR) of 82.60% on classification tasks and significant improvements over baselines on captioning and retrieval benchmarks. The framework's ability to disrupt semantic-level alignment rather than merely introducing visual artifacts enables superior cross-model transferability while maintaining computational efficiency.

**Compliance With Llm Reviewing Policy:**

Affirmed.

**Final Justification:**

My concerns have been addressed, so I am maintaining my positive score.

**Key Questions For Authors:**

See Weaknesses

**Limitations:**

yes

**Strengths And Weaknesses:**

Strengths

1. MADA-Attack introduces the first comprehensive framework for text-guided non-targeted UAPs in VLMs, moving beyond unimodal visual attacks to explicitly target cross-modal semantic alignment.
2. The consistent SOTA performance demonstrated in both intra-dataset and cross-dataset settings strongly validates the method's effectiveness.
3. The work is grounded in rigorous analysis of cross-modal attention mechanisms and optimization dynamics, with valuable insights into the "disrupt-then-exploit" pattern in VLM vulnerabilities.

Weaknesses

1. The method shows reduced effectiveness on certain tasks like VizWiz-VQA, where answer distribution skew leads to evaluation bias. The paper acknowledges this issue but doesn't provide adequate solutions.
2. There's no analysis of computational overhead during inference, real-time attack feasibility, or robustness against common defenses like input purification or adversarial training.

---

> ### Author Rebuttal · Authors · 2026-03-30
>
> We sincerely appreciate your constructive suggestions, valuable review and recognition of our work. Below are our responses to your questions.
>
> ---
> **Q1**: Adequate solutions for reduced effectiveness on VQA tasks.
> **A1**: We would like to clarify the underlying cause. Our method is trained on datasets that largely excluding unanswerable or uncertain visual–text pairs, whereas VizWiz-VQA contains a substantial portion of such cases. As highlighted in recent study, VLMs exhibit a “**hallucination**” tendency, i.e., they favor semantically plausible answers even when visual evidence is insufficient [1]. The absence of these uncertain cases leads to a mismatch in optimization, as our attack does not explicitly account for such outlier scenarios.
>
> Moreover, we note an additional evaluation-level factor. Answers in VizWiz-VQA are often represented as a set of possible responses. Although the UAP can **successfully alter** the model’s semantic interpretation, the predicted answer may still overlap with elements in the ground-truth set, thereby underestimating the true attack success and resulting in a lower observed ASR.
>
> We acknowledge that the paper does not fully resolve this issue, and we will extend the discussion by outlining potential directions for improvement:
> -  Incorporating uncertain and **unanswerable samples** to better capture out-of-distribution characteristics.
> - Designing **uncertainty-aware objectives** to mitigate hallucination effects.
> We believe these directions can inspire more robust future designs, and we will include a detailed discussion in the revised manuscript.
>
> *[1] Seeing but Not Believing: Probing the Disconnect Between Visual Attention and Answer Correctness in VLMs, ICLR 2026*
>
> ---
> **Q2**: Overhead, feasibility of real-time attack and robustness of *MADA-Attack*.
> **A2**: (1)**Overhead**. *MADA-Attack* leverages textual perturbations only during the **training phase** to guide the optimization of image perturbations, enabling a text-guided non-targeted adversarial attack (see Section 1 and Eq. 4). During inference, the attack introduces **negligible computational overhead**, as it only requires adding a pre-computed perturbation to the input image, without any additional model interaction or text processing.
>
> **(2)Real-time attack**. Based on the analysis above, *MADA-Attack* is designed for transferable attacks under black-box settings, which naturally supports **real-time deployment**.
>
> **(3)Robustness**. Regarding **robustness against defenses**, we have included a detailed evaluation and analysis against adversarial trained encoders such as $TeCoA^2$ and $Fare^2$ (see Appendix G). We note that adversarial training provides some degree of robustness, as reflected by an decreased average ASR from 82.60% to 15.63%, vs. X-Transfer 70.83 to 12.37, but such improvement remains limited in adversarial training forms of **$L_\infty$**.
>
> Specifically, most adversarial training strategies are designed around $L_\infty$​-bounded perturbations, which effectively suppress pixel-level UAPs such as X-Transfer. While this leads to noticeable robustness improvements, it also implicitly constrains the defense to a particular class of perturbations. This observation is further supported by prior work, which shows that patch-based perturbations can bypass $L_\infty$​-trained defenses, highlighting that adversarial robustness is **highly dependent on the perturbation form** ([1]).
>
> While these properties are implicitly reflected in the current manuscript, we acknowledge that their presentation may not be sufficiently explicit. We will revise the paper to more clearly highlight the efficiency and practical feasibility of *MADA-Attack* for improved clarity.
>
> *[1] X-Transfer Attacks: Towards Super Transferable Adversarial Attacks on CLIP, ICML 2025*

---

> > ### Author Rebuttal · Reviewer_MFYv · 2026-04-01
> >
> > My concerns have been adequately addressed.

---

> > > ### Author Response · Authors · 2026-04-02
> > >
> > > We sincerely thank you for your positive feedback and for acknowledging that your concerns have been fully addressed. We also greatly appreciate your insightful comments regarding evaluation bias, computational overhead, and robustness. These suggestions are highly valuable and have helped us further improve the clarity of the paper, as well as identify important directions for future work.

---

### Decision · Program_Chairs · 2026-04-30

**Decision:**

Accept (regular)

**Comment:**

Overall, the paper proposes a well-motivated multimodal adversarial attack framework (MADA-Attack) and demonstrates strong empirical performance across multiple tasks and models. Reviewers generally agree that the method effectively extends UAP attacks from unimodal to multimodal settings. In particular, reviewers MFYv and nsnL highlight the clear motivation and solid experimental validation. They confirm that their concerns were fully addressed after rebuttal. However, several reviewers (e.g., kYb4 and UYU4) point out that the technical novelty is relatively incremental, mainly combining existing components without introducing fundamentally new insights and insufficient theoretical analysis. Despite these concerns, most reviewers maintain weak accept scores after rebuttal, suggesting that the work is technically sound and useful for the community.